# Intrinsic control of muscle attachment sites matching

Alexandre Carayon[1,2,3], Laetitia Bataillé[1,2,3], Gaëlle Lebreton[1,2,3], Laurence Dubois[1,2,3], Aurore Pelletier[1,2,3], Yannick Carrier[1,2,3], Antoine Wystrach[2,3], Alain Vincent[1,2,3], Jean-Louis Frendo[1,2,3]*

[1]Centre de Biologie du Développement (CBD), Toulouse, France; [2]Centre de Recherche sur la Cognition Animale (CRCA), Toulouse, France; [3]Centre de Biologie Intégrative (CBI), Université de Toulouse, CNRS, UPS, Toulouse, France

**Abstract** Myogenesis is an evolutionarily conserved process. Little known, however, is how the morphology of each muscle is determined, such that movements relying upon contraction of many muscles are both precise and coordinated. Each *Drosophila* larval muscle is a single multinucleated fibre whose morphology reflects expression of distinctive identity Transcription Factors (iTFs). By deleting transcription cis-regulatory modules of one iTF, Collier, we generated viable muscle identity mutants, allowing live imaging and locomotion assays. We show that both selection of muscle attachment sites and muscle/muscle matching is intrinsic to muscle identity and requires transcriptional reprogramming of syncytial nuclei. Live-imaging shows that the staggered muscle pattern involves attraction to tendon cells and heterotypic muscle-muscle adhesion. Unbalance leads to formation of branched muscles, and this correlates with locomotor behavior deficit. Thus, engineering *Drosophila* muscle identity mutants allows to investigate, in vivo, physiological and mechanical properties of abnormal muscles.

*For correspondence:
jean-louis.frendo@univ-tlse3.fr

Competing interests: The authors declare that no competing interests exist.

## Introduction

The musculature of each animal is composed of an array of body wall muscles allowing precision and stereotypy of movements. The somatic musculature of the *Drosophila* larva - about 30 distinct body wall muscles per each hemi-segment which are distributed in three layers, internal, median and external - is a model to study how a muscle pattern is specified and linked to locomotion behaviour. Each muscle is a single multinucleated fibre with a specific identity: size, shape, orientation relative to the dorso-ventral and antero-posterior body axes, motoneuron innervation and attachment sites to the exoskeleton via tendon cells located at specific positions. Intersegmental tendon cells, where a large fraction of muscles is attached, are distributed in three groups, dorsal, lateral and ventral (*Bate, 1990*; *Volk and VijayRaghavan, 1994*; *Schweitzer et al., 2010*; *Armand et al., 1994*). Internal muscles also attach to muscle(s) in the next segment, forming 'indirect' muscle attachment sites (iMAS) (*Maartens and Brown, 2015*).

*Drosophila* muscle development proceeds through fusion of a founder myoblast (founder cell, FC) with fusion-competent myoblasts (FCMs) (*Deng et al., 2017*). FCs originate from asymmetric division of progenitor cells (PCs) themselves selected from equivalence groups of myoblasts, called promuscular clusters (PMCs). Muscle identity ensues from the expression by each PC and FC of a specific combination of identity transcription factors (iTFs) (*de Joussineau et al., 2012*), established in three steps: First, different iTFs are activated in different PMCs, in response to positional information, and this expression is only maintained in PCs. Second, refinement of the iTF code occurs via cross-regulations between different iTFs in PCs and/or FCs and Notch signalling (*Carmena et al., 1998*; *Carmena et al., 2002*; *Enriquez et al., 2010*). Several PCs can be serially selected from a PMC and give rise to different muscle identities according to birth order, adding a temporal

**eLife digest** Each muscle in the body has a unique size, shape and set of attachment points. Animals need all of their muscles to have the correct identity to help maintain posture and control movement. A specific set of proteins, called transcription factors, co-ordinate and regulate gene activity in cells so that each muscle develops in the right way.

To create a muscle, multiple precursor cells fuse together to form a muscle fibre, which then elongates and attaches to specific sites. Correct attachment is critical so that the fibre is properly oriented. When this process goes wrong, for example in disease, muscle fibres sometimes attach to the wrong site; they become branched and cannot work properly.

Collier is a transcription factor protein that controls muscle identity in the fruit fly *Drosophila melanogaster.* However, like many transcription factors, Collier also has several other roles throughout the body. This made it difficult to evaluate the effect of the protein on the formation of specific muscles.

Here, Carayon et al. managed to selectively deactivate Collier in just one muscle per body section in the larvae of fruit flies. This showed that the transcription factor is needed throughout muscle development; in particular, it is required for muscle fibres to select the correct attachment sites, and to be properly oriented. Affected muscles showed an altered orientation, with branched fibres attaching to the wrong site. Even minor changes, which only affect a single muscle from each body segment, greatly impaired the movement of the larvae.

The work by Carayon et al. offers a new approach to the study of muscular conditions. Branched muscles are seen in severe human illnesses such as Duchenne muscular dystrophy. Studying the impact of these changes in a living animal could help to understand how this disease progress, and how it can be prevented.

dimension to these regulations. A well-documented example is the distinction between DA2 and DA3 identities (*Figure 1A*; *Dubois et al., 2016*; *Boukhatmi et al., 2012*). Third, transcriptional activation of iTFs in syncytial nuclei after fusion correlates with the activation of identity 'realisation' genes acting downstream of some iTFs (*Crozatier and Vincent, 1999*; *Knirr et al., 1999*; *Bataillé et al., 2010*; *Bataillé et al., 2017*).

The consequence of specific muscle identity defects on locomotion, a question of prime importance for progress on studying human myopathies which affect subsets of muscles, remains largely to be assessed. Genetically controlled muscle identity changes should, in principle, provide suitable models for studying locomotion deficits linked to muscle imbalance. However, mutations for known *Drosophila* iTFs are embryonic lethal and/or show pleiotropic phenotypes reflecting their multiple expression sites. Here, we took advantage of our previous characterisation of *col* expression in a single larval muscle, the DA3 muscle and of the two involved cis-regulatory modules (CRMs), early (E-CRM) and late (L-CRM), to generate muscle-specific mutants and circumvent lethality/pleiotropy of null mutants.

CRM deletions show that E-CRM and L-CRM act redundantly at the PC stage, emphasising that PC is a key stage in specification of muscle identity. L-CRM deletion results into loss of *col* transcription in DA3 FCs and morphological transformation of the DA3 into a DA2-like muscle. Removal of an auto-regulatory cis-module located in L-CRM specifically abolishes *col* activation in syncytial nuclei fusing with the DA3 FC. This leads to incomplete DA3 transformations and the formation of bifid/branched muscles of mixed DA3/DA2 morphology. In summary, our data show that i) the FC transcriptional program must be propagated to syncytial nuclei for a muscle to adopt a specific morphology; ii) the precise matching of muscle/muscle attachments over the intersegmental border, which leads to a staggered rows pattern, involves a process of selective adhesion controlled by iTFs; iii) branched muscles affect larval locomotion performance.

Branched muscles are typical of late, severe phases of human Duchenne Muscular Dystrophy (*Chan and Head, 2011*). *Drosophila* iTF CRM deletion could be an effective setting for creating muscle-specific transformations and branched muscles, as new paradigms to study myopathies specifically affecting subsets of muscles in humans.

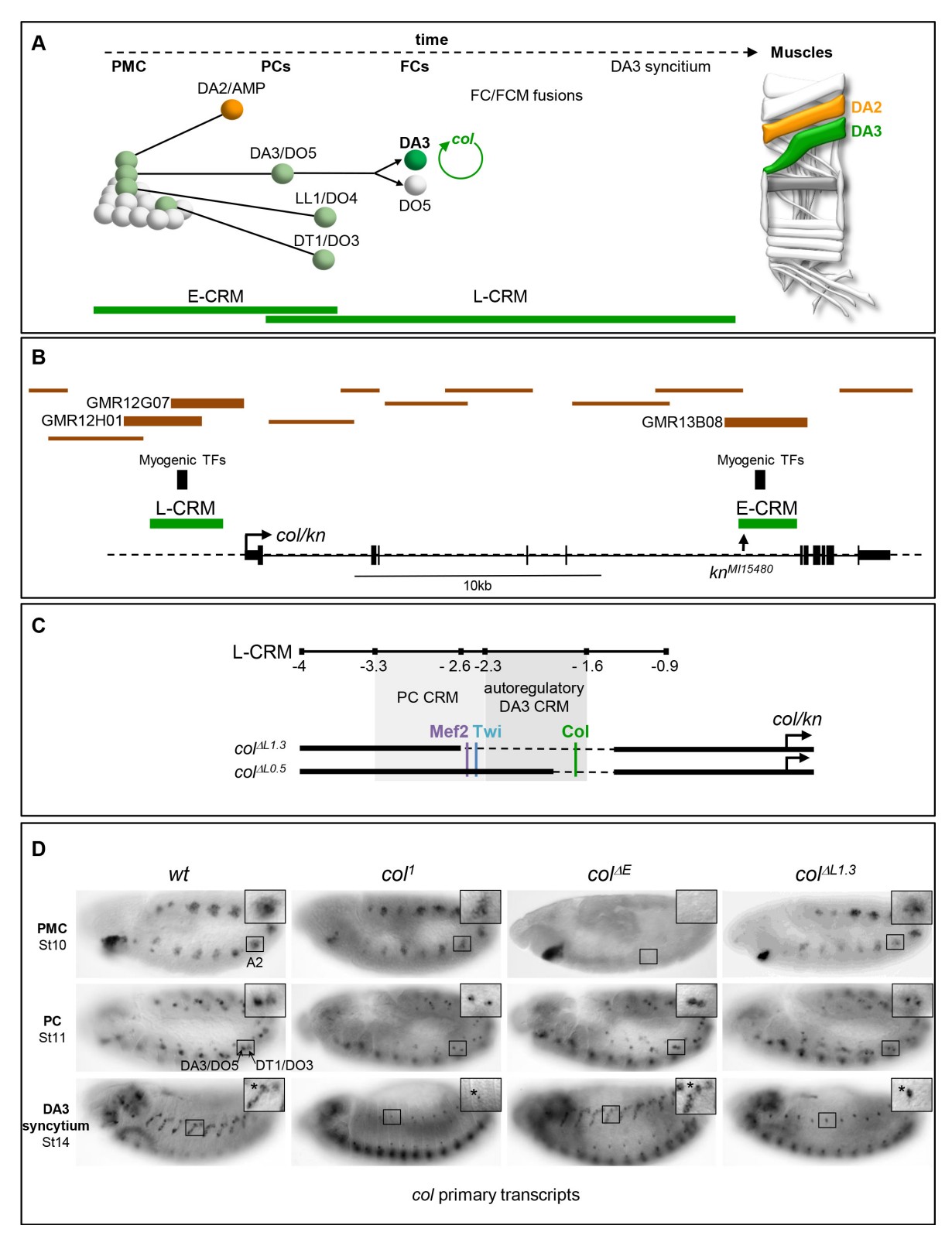

**Figure 1.** *Col* CRMs, CRM deletions and *col* transcription. (**A**) Diagrammatic representation of the sequential emergence of 4 different PCs from the Col-expressing PMC (grey), division of the DA3/DO5 PC into 2 FCs and *col* auto-regulation in the DA3 lineage; the names of each PC and FC are indicated. Accumulation of Col protein is in green. The time windows of mesodermal early (E-CRM) and late (L-CRM) CRM activity are indicated by green lines. Right: muscle pattern of an abdominal segment highlighting DA2 (orange) and DA3 (green). (**B**) Schematic representation of the *col*
*Figure 1 continued on next page*

*Figure 1 continued*

transcribed region (http://flybase.org/cgi-bin/gbrowse2/dmel/?Search=1;name=FBgn0001319). The position of tested GMR and VT fragments are drawn as brown horizontal bars; the numbers are given for those active in DA3. The positions of clusters of in vivo Mef-2, Twi and Tin binding sites are indicated by vertical blue bars, the $kn^{Mi15480}$ transposon insertion, used for the E-CRM deletion screening, by a vertical arrow. (C) Enlarged view of L-CRM indicating the juxtaposition of PC-specific and autoregulatory DA3-specific CRMs, Col, Mef2 and Twi binding sites, and the $col^{\Delta L1.3}$ and $col^{\Delta L0.5}$ deletions generated by CRISPR/cas9 genome editing. (D) *Col* transcription in wt and mutant embryos, genotypes indicated, visualised by in situ hybridisation to primary transcripts. A detail of the abdominal A2 segment (squared area) is shown in each panel. PMC *col* transcription is lost in $col^{\Delta E}$ embryos; stage 11, *col* transcription in the DA3/DO5 and DT1/DO4 PCs is detected in all strains; stage 14, DA3 syncytium transcription is lost in $col^1$ and $col^{\Delta L1.3}$ embryos. * indicates col transcription in a multidendritic neuron (md).

The online version of this article includes the following figure supplement(s) for figure 1:

**Figure supplement 1.** Genomic sequence of the E-CRM deletion.
**Figure supplement 2.** Genomic sequence of the L-CRM deletions.
**Figure supplement 3.** Mapping of a PC-specific CRM.

## Results

Redundant CRMs at the PC step ensure robustness of iTF transcription *col* transcription in myogenic lineages is first observed in a dorso-lateral PMC from which are sequentially selected several PCs (*Dubois et al., 2016*). It is subsequently maintained in two PCs, then one FC, the DA3 FC, and is activated in FCM nuclei recruited into the DA3 growing fibre (*Crozatier and Vincent, 1999*; *Figure 1A*). Two *col* CRMs containing embryonic in vivo binding sites for the master myogenic TFs, Mef-2 and Twist (Twi) (*Sandmann et al., 2007*; *Zinzen et al., 2009*) were previously identified, which reproduce this sequence of expression in reporter assays: E-CRM, which is active in the PMC and the DA3/DO5 and DT1/DO3 PCs, and L-CRM which is active in these same 2 PCs, the DA3 FC and DA3 syncytial nuclei (*Enriquez et al., 2010*; *Dubois et al., 2007*; *Figure 1A*; *Figure 1—figure supplement 1* and *Figure 1—figure supplement 2*). With the prospect of using CRM deletions to generate muscle-specific mutants and to exclude the possible existence of additional, redundant enhancers (*Cannavò et al., 2016*), we conducted a systematic analysis of Gal4 reporter lines (*Manning et al., 2012*; *Kvon et al., 2014*) covering 36 kb of the *col* genomic region. A single reporter, GMR13B08, reproduces PMC Col expression and it overlaps E-CRM, and two reporters, GMR12G07 and GMR12H01, display DA3 expression and they both partly overlap L-CRM (*Figure 1B* and data not shown), attesting to the existence of only two *col* muscle CRMs.

The activity of E-CRM and L-CRM at different phases of muscle development raised the question of their respective roles in defining muscle identity. To address this question, we separately deleted each of them from the genome using the CrispR/Cas9 technology. Deletion of a 2,4 kb fragment removed the E-CRM. Two deletions within the L-CRM were generated: deletion of a 1.3 kb fragment removes the entire core region (*Dubois et al., 2007*) and deletion of a 0.5 kb fragment removes the Col autoregulation site (*de Taffin et al., 2015*; *Figure 1C*; *Figure 1—figure supplement 1* and *Figure 1—figure supplement 2*). The corresponding mutant *Drosophila* strains are designated $col^{\Delta E}$, $col^{\Delta L1.3}$ and $col^{\Delta L0.5}$, respectively. $col^{\Delta L}$ strains are homozygous viable and fertile. $col^{\Delta E}$ strain is homozygous female sterile, a sterility unrelated to *col* activity, since fertility is restored by placing $col^{\Delta E}$ over a deficiency (*Df(2L)BSC429*) (abbreviated *Df* in the rest of the text and figures) removing the entire *col* locus (data not shown). As a first step to determine the consequences of deleting either E-CRM or L-CRM, we compared *col* transcription between wt, $col^1$ (a protein null mutant), $col^{\Delta E}$ and $col^{\Delta L1.3}$ strains, using an intronic probe to detect nascent transcripts (*Figure 1D*). In $col^1$ homozygous mutant embryos, *col* transcription is detected at the PMC and PC stages and lost at the FC stage, showing the key role of autoregulation in the maintenance of col transcription (*Crozatier and Vincent, 1999*; *de Taffin et al., 2015*). In $col^{\Delta E}$ embryos, we found that *col* is not activated in PMC cells, as could be expected. Yet, *col* transcription is detected in the DA3/DO5 and DT1/DO3 PCs, showing that inheritance of Col protein synthesised under E-CRM control is not required for activation of *col* transcription in PCs. A normal pattern is observed in the developing DA3 muscle, at stage 14. Reciprocal to E-CRM deletion, *col* is transcribed in PMC cells and the DA3/DO5 and DT1/DO3 PCs in $col^{\Delta L1.3}$ embryos, but no more at stage 14, showing that L-CRM is required for *col* transcription maintenance.

Both E-CRM and L-CRM activity are detected in PCs. In absence of E-CRM, no Col protein is inherited from the PMC. Therefore, L-CRM activity in PCs cannot be due to *col* autoregulation.

Analysis of new *col-lacZ^yi* reporter genes (*Perry et al., 2010*) revealed the existence of a PC-specific CRM located within the −3.3 to −2.3 fragment of L-CRM, *i.e.*, which is separate from the autoregulatory CRM defined by the in vivo Col binding site and contains the in vivo binding sites for Mef2 and Twist (*Figure 1C*; *Figure 1—figure supplement 3*; *Zinzen et al., 2009*). Activity of this PC-specific CRM is transient and lost upon removal of Mef2 and Twist binding sites (*Figure 1—figure supplement 3*). Overall, we conclude that two CRMs separately drive *col* transcription in muscle PCs, suggesting that robust iTF expression at the PC stage is critical to confer a muscle its identity.

## *col* CRM deletions lead to muscle transformation and branched muscles

Having deleted separately each *col* CRM allowed to assess the respective roles of iTF transcription before or after the PC step. To compare the different CRM deletions, we placed each of them over the deficiency Df chromosome. We introduced the L-CRM-moeGFP reporter to visualise the DA3 morphology at stage 15 (*Enriquez et al., 2012*). Control (+/*Df*) embryos display reporter expression

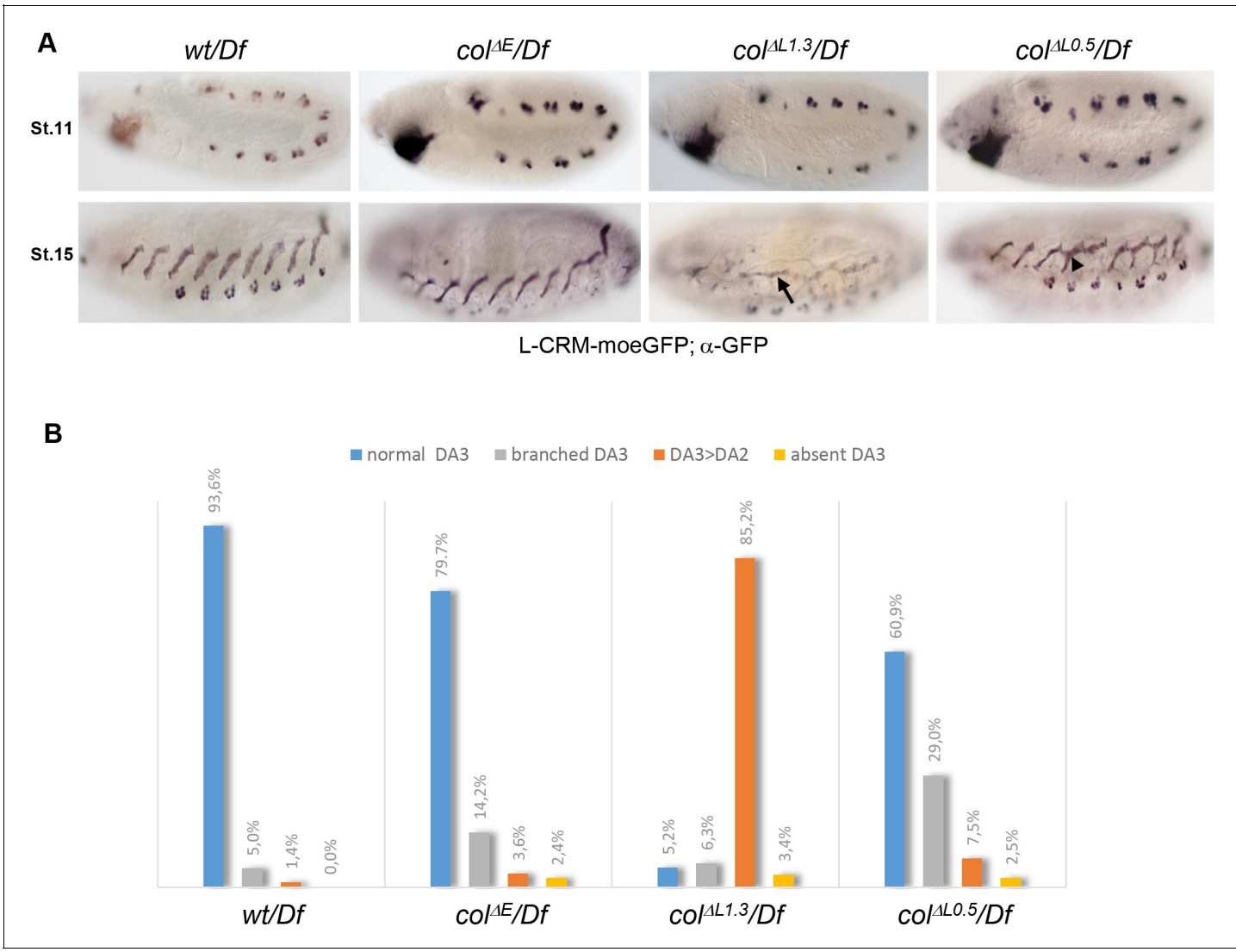

**Figure 2.** DA3 muscle transformations upon *col-CRM* deletions. (**A**) L-CRM-moeGFP expression in stage 11 and 15 hemizygous embryos, as indicated. GFP expression in PCs at stage 11 is similar in all strains. DA3^>DA2 transformations (arrow) and branched DA3 muscles (arrowhead) are observed in *col^ΔL* embryos. (**B**) Quantification of the relative proportions of normal DA3, branched DA3, DA3^>DA2 transformation and absence of DA3 muscles in *wt*, *col^ΔE*, *col^ΔL1.3* and *col^ΔL0.5* hemizygous embryos. A minimum of 100 A1-A7 abdominal segments of stage 15–16 embryos were analysed for each genotype. (+/Df: n = 127 segments - 16 embryos; *col^ΔE*/Df: n = 170 segments - 23 embryos; *col^ΔL1.3*/Df: n = 190 segments - 27 embryos; *col^ΔL0.5*/Df: n = 103 segments - 13 embryos).

in the DA3/DO5 and DT1/DO3 PCs at stage 11 and the DA3 muscle at stage 15 (*Figure 2A*). The same pattern is observed in PCs for all *col*-CRM deletion strains, consistent with transcript analyses (*Figure 1D*). L-CRM-moeGFP expression at stage 15 shows that the DA3 morphology is normal in about 80% of segments in $col^{\Delta E}/Df$ embryos (*Figure 2A–B*), and we did not pursue the analysis of this deletion strain. Low level GFP expression in $col^{\Delta L1.3}/Df$ embryos, consistent with *col* transcription data (*Figure 1D*), shows that the DA3 muscle is most often (85.2% of segments) transformed into a DA2-like muscle (designated below as $DA3^{>DA2}$; *Figure 2A–B*), like in *col* null mutant embryos (*Enriquez et al., 2012*). In $col^{\Delta L0.5}/Df$ embryos, *i.e*, when only the autoregulation module has been deleted, a high number (29%) of branched muscles is observed (*Figure 2A–B*). Branched muscles correspond to incomplete transformations, with two stable anterior attachment sites, overlapping the DA3 and DA2 sites in *wt* embryos. The high ratios of either complete ($DA3^{>DA2}$) or incomplete (branched) transformations in L-CRM deletion mutants demonstrate that an iTF CRM deletion strategy is effective for creating viable muscle-specific identity mutants and explore branched muscle properties.

## Re-programming of syncytial nuclei is required for muscle morphological identity

Complete *vs* incomplete transformations in $col^{\Delta L1.3}$ versus $col^{\Delta L0.5}$ deletions suggest that proper level and/or maintenance of iTF expression is crucial for proper muscle development. This led us to compare the pattern of Col protein in growing DA3 syncytium between *wt*, $col^{\Delta L1.3}$ and $col^{\Delta L0.5}$ embryos. In either deletion strain, Col is detected in PCs at stage 11 but not in muscles at stage 15 (*Figure 3A*). However, at stage 14, some Col protein is still detected in muscle precursors in $col^{\Delta L0.5}$, not in $col^{\Delta L1.3}$ embryos (*Figure 3B*). To trace the origin of this difference, we examined *col* transcription in the DA3 PC, FC and stage 14 syncytium using *Df*/hemizygous embryos which display one hybridisation dot per active nucleus (*Figure 3C*). In control *wt/Df* embryos, a dot is systematically detected in the DA3/DO5 PC (20/20 segments; five embryos analysed), the DA3 FC (20/20) and 80% of DA3 nuclei at stage 14 (6 of 7–8 nuclei per fibre on average; 27 segments). A dot is detected as well in the DA3/DO5 PC, in either $col^{\Delta L1.3}$ (21/21) or $col^{\Delta L0.5}$ (18/18) embryos, reflecting E-CRM activity (*Figure 1D*). In $col^{\Delta L0.5}$ embryos, a *col* hybridisation dot is detected in the DA3 FC (19/19) and in one nucleus, likely the FC nucleus (11/21 segments), sometimes two nuclei at stage 14. In $col^{\Delta L1.3}$ embryos, however, *col* transcription is only detected in a minor fraction of FCs (4/15) and is completely lost at stage 14 (0/6–7 nuclei per fibre on average; 24 segments). Patterns similar to stage 14 are observed at stage 15, while at stage 16, *col* transcription is detected neither in L-CRM deletion strains nor in control (*Figure 3—figure supplement 1*). Since *col* transcription at, and from, the PC stage appears to be nodal to DA3 identity, we measured the level of *col* transcripts relative to *nautilus* (*nau*), the *Drosophila MyoD-MRF* serving as internal reference (*Figure 3D* and *Figure 3—figure supplement 2*). As expected, similar levels of *col* transcription are found in the DA3 PC, in control (Mean ± sem: 1.22 ± 0.06; n = 18), $col^{\Delta L1.3}$ (1.15 ± 0.04; n = 26) and $col^{\Delta L0.5}$ embryos (1.19 ± 0.04; n = 18), which confirms handling by the E-CRM, with only a minor contribution from the L-CRM (*Figure 1D*). On the contrary, high level of *col* transcription in the DA3 FC in wt embryos (2.20 ± 0.05; n = 28) is dependent upon L-CRM activity, since a drop is observed in $col^{\Delta L1.3}$ FCs (1.10 ± 0.04; n = 24), p<0.001. More precisely, it is dependent upon the presence of Mef2 and Twist binding sites since a basal level of transcription is still observed when only the *col* autoregulation module is deleted ($col^{\Delta L0.5}$: 1.41 ± 0.04; n = 25). Quantification of *col* and *nau* transcripts shows that the *col/nau* increase between the PC and FC stages in wt embryo is due to increased *col* transcription while the level of *nau* is relatively constant and is unaffected in *col* L-CRM mutants (*Figure 3—figure supplement 2*). Together, the data suggest that binding of Mef2 and Twist is required to prime *col* transcription in the FC nucleus before autoregulation takes off (*Figure 3E*). Taken with one another, Col immunostaining, FISH data and *col* transcription quantification indicate that sustained *col* transcription in the FC nucleus in $col^{\Delta L0.5}$ embryos (*Figure 3C–D*), provides enough Col protein for some uptake by other DA3 nuclei at stage 14, and this leads to their partial reprogramming to DA3 identity. Partial reprogramming could, in turn, explain the formation of branched muscles retaining some DA3 morphological characters. Moreover, the $col^{\Delta L1.3}$ and $col^{\Delta L0.5}$ expression data and deletion phenotypes show that both iTF transcription in the FC and reprogramming of 'naïve' syncytial nuclei contribute to ensure robust muscle morphological identity (*Figure 3E*).

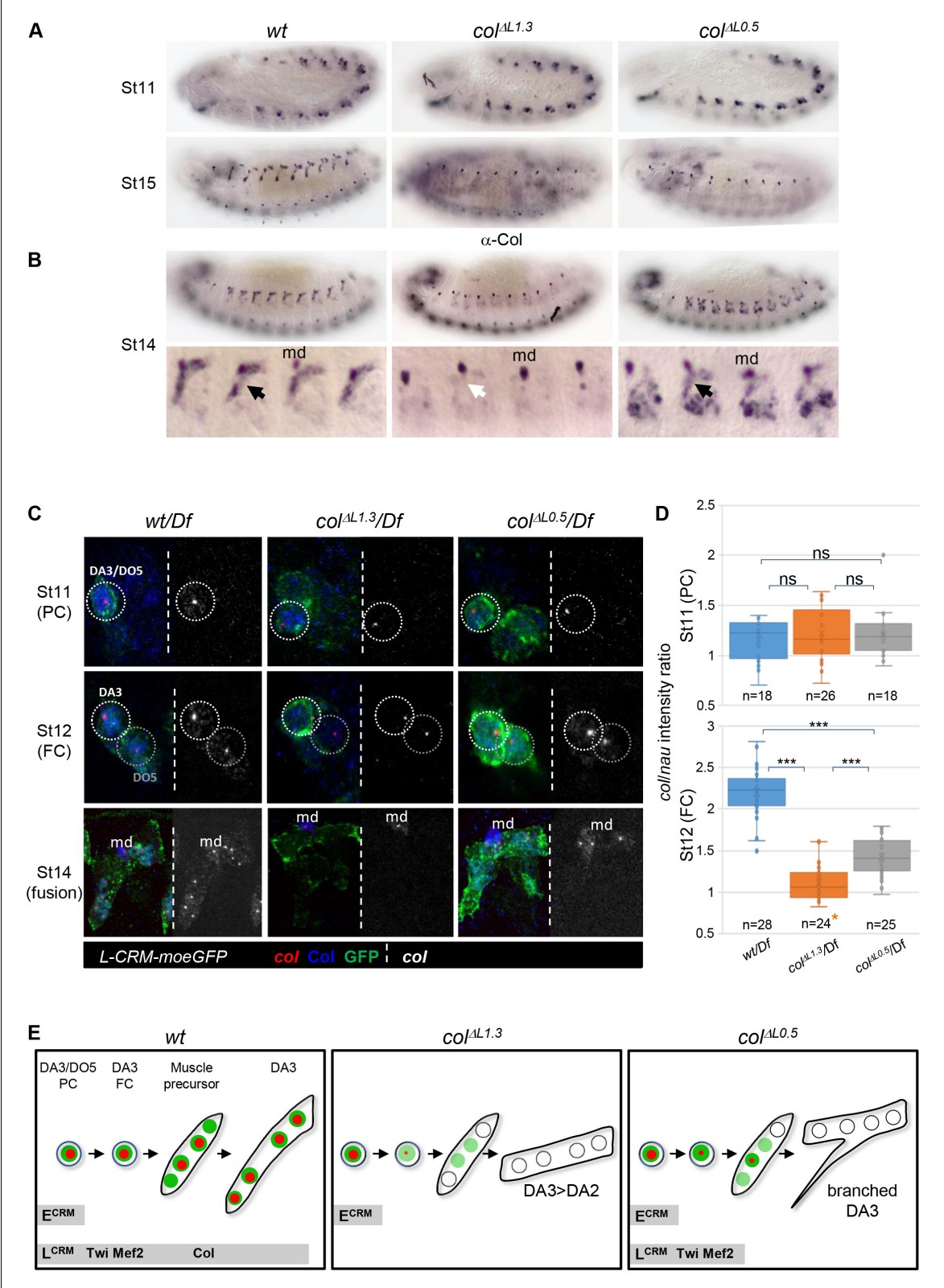

**Figure 3.** Identity reprogramming of syncytial nuclei controls the final muscle morphology. (A) Col immunostaining of wt, $col^{\Delta L1.3}$ and $col^{\Delta L0.5}$ embryos, showing a normal pattern at stage 11 and complete absence of Col protein at stage 15 in both $col^{\Delta L}$ embryos. (B) Stage14 embryos with a close up view of 4 segments shows low amounts of Col protein (black arrow) in the growing DA3 muscle in $col^{\Delta L0.5}$ and absence in $col^{\Delta L1.3}$ embryos (white arrow). Position of the multidendritic neuron (md) is indicated. (C) Col transcription (red dots), Col protein (blue) and L-CRM-moeGFP expression

*Figure 3 continued on next page*

*Figure 3 continued*

(green) in the DA3/DO5 PC, stage 11, DA3 FC, stage 12, and developing DA3 muscle, stage 14, in *wt/Df*, *col*$^{\Delta L1.3}$/*Df* and *col*$^{\Delta L0.5}$/*Df*; L-CRM-moeGFP embryos. In each panel, *col* transcripts are shown separately in black and white. *col* transcription ceases after FC stage in *col*$^{\Delta L1.3}$ embryos and does not propagate to other syncytial nuclei in *col*$^{\Delta L0.5}$ embryos. (D) Quantification of col primary transcripts level in PC and FC nuclei; orange asterisk: *col* transcription in FCs is generally lost in *col*$^{\Delta L1.3}$ embryos; quantification was done on a small fraction of FCs; n: number of PC or FC analysed, using 5 or six embryos at each stage 11 and 12; 15 *col*$^{\Delta L1.3}$ embryos were used for the FC stage (Mean ± sem and ***: p<0.001). (E) Schematic representation of the dynamics of *col* transcription (red dots) and Col protein (green) in the DA3/DO5 PC, the DA3 FC, muscle precursor, and DA3, DA3$^{>DA2}$ and branched DA3 muscles in wt, *col*$^{\Delta L1.3}$ and *col*$^{\Delta L0.5}$ embryos, respectively. Temporal activity of E-CRM and L-CRM is represented by horizontal grey bars. The online version of this article includes the following figure supplement(s) for figure 3:

**Figure supplement 1.** Col transcription in *col*$^{\Delta L0.5}$ and *col*$^{\Delta L1.3}$ embryos In situ hybridisation to *col* primary transcripts in stage 15 and 16 *wt/Df*, *col*$^{\Delta L1.3}$/*Df* and *col*$^{\Delta L0.5}$/*Df* embryos.

**Figure supplement 2.** *Nau* expression is not affected in *col*$^{\Delta L0.5}$ and *col*$^{\Delta L1.3}$ embryos.

## Muscle attachment: tendon attraction and muscle-muscle matching

We next investigated in more detail how ectopic muscle attachment sites at the origin of transformed and branched muscles are selected and stabilised. Double staining of ready-to-hatch (st17) *wt* embryos for F-actin and Ilk-GFP, a component of muscle attachment sites (*Zervas et al., 2011*; *Sarov et al., 2016*) shows that the DA3 posterior edge anchors to dorsal, and anterior edge to lateral intersegmental tendon cells (*Figure 4A*), giving its final acute shape (*Bate, 1990*). Moreover, the DA3 and DA2, as well as the DA2 and DA1 muscles precisely align with each other over each intersegmental border, forming heterotypic muscle-muscle attachment (iMAS; [*Maartens and Brown, 2015*]) at the origin of the staggered rows disposition of DA muscles. No DA3/DA3 (homotypic) iMAS surface is observed (*Figure 4A*). On the contrary, in *col*$^{\Delta L1.3}$ embryos, the anterior edge of DA3$^{>DA2}$ transformed muscles anchors to dorsal tendon cells instead of lateral intersegmental tendon cells, leading to a 'dual DA3$^{>DA2}$ morphology', DA2-like at the anterior, and DA3 at the posterior edge. This dual identity leads to iMASs between adjacent DA3$^{>DA2}$ muscles (*Figure 4A*). As a consequence, DA2 iMASs are shifted dorsally and the general pattern of DA muscles is affected.

To explore the dynamics of DA3 muscle attachment, we live-imaged *wt* and *col*$^{\Delta L1.3}$ embryos, starting at stage 12 (defined as t$_0$ in *Videos 1* and *2*). The DA3 muscle is visualised by L-CRM-moeGFP and tendon cell along the entire intersegmental border by *stripe* Gal4;UASmCD8RFP (*Volohonsky et al., 2007*). In *wt* embryos between stages 12 and 13 (*Video 1*; *Figure 4B*), the DA3 muscle precursor extends protrusions towards dorsal tendon cells, posteriorly, and both dorsal and lateral tendon cells, anteriorly, diverging from a bipolar extension scheme (*Schnorrer and Dickson, 2004*). At stage 14, the posterior DA3 edge makes contacts with dorsal tendon cells before the anterior edge(s) reaches the intersegmental border, a time gap previously observed during live-imaging of a ventro-lateral muscle (*Gilsohn and Volk, 2010*). In addition, numerous filopodia emanate from the DA3 dorsal surface and contact the DA2 ventral surface, contributing to give the DA3 muscle precursor its transient angled-shape. At late stage 14, the posterior DA3 attachment widens parallel to the intersegmental border. Its anterior attachment to lateral tendon cells in turn becomes stable, whereas dorsal anterior filipodia appear to be repelled from the intersegmental border where the DA3 muscle in the preceding segment is attached. Remaining filopodia are limited to the rims of DA3 anchoring, possibly suggesting a mechanism of homotypic repulsion. In *col*$^{\Delta L1.3}$ embryos, the main axis of the DA3$^{>DA2}$ muscle precursor elongation is more longitudinal than *wt*, already at stage 12 (*Video 2*; *Figure 4B*). At stage 13 and like in *wt*, many protrusions emanate from the DA3 dorsal surface, but unlike in *wt*, protrusions contacting dorsal tendon cells do not retract at later stages when they contact the DA3$^{>DA2}$ muscle from the preceding segment. Rather, stabilisation of contacts made by these protrusions prefigures abnormal DA3$^{>DA2}$/DA3$^{>DA2}$ iMAS formation (*Figure 4A*). In some segments, contacts with lateral tendon cells are also stabilised, resulting into branched muscles. In summary, imaging of live *wt* embryos illustrates different steps involved in establishment of the acute DA3 orientation: posterior attachment to dorsal tendon cells at the same time as anterior exploration of dorsal and lateral tendon cell. This is followed by DA3 attachment to lateral, and retraction from dorsal tendon cells (*Figure 4B*, *Video 1*), cumulating into heterotypic DA3/DA2 iMAS stabilisation (*Figure 4A and B*). This retraction does not take place in *col*$^{\Delta L1.3}$ embryos (*Figure 4B* and *Video 2*).

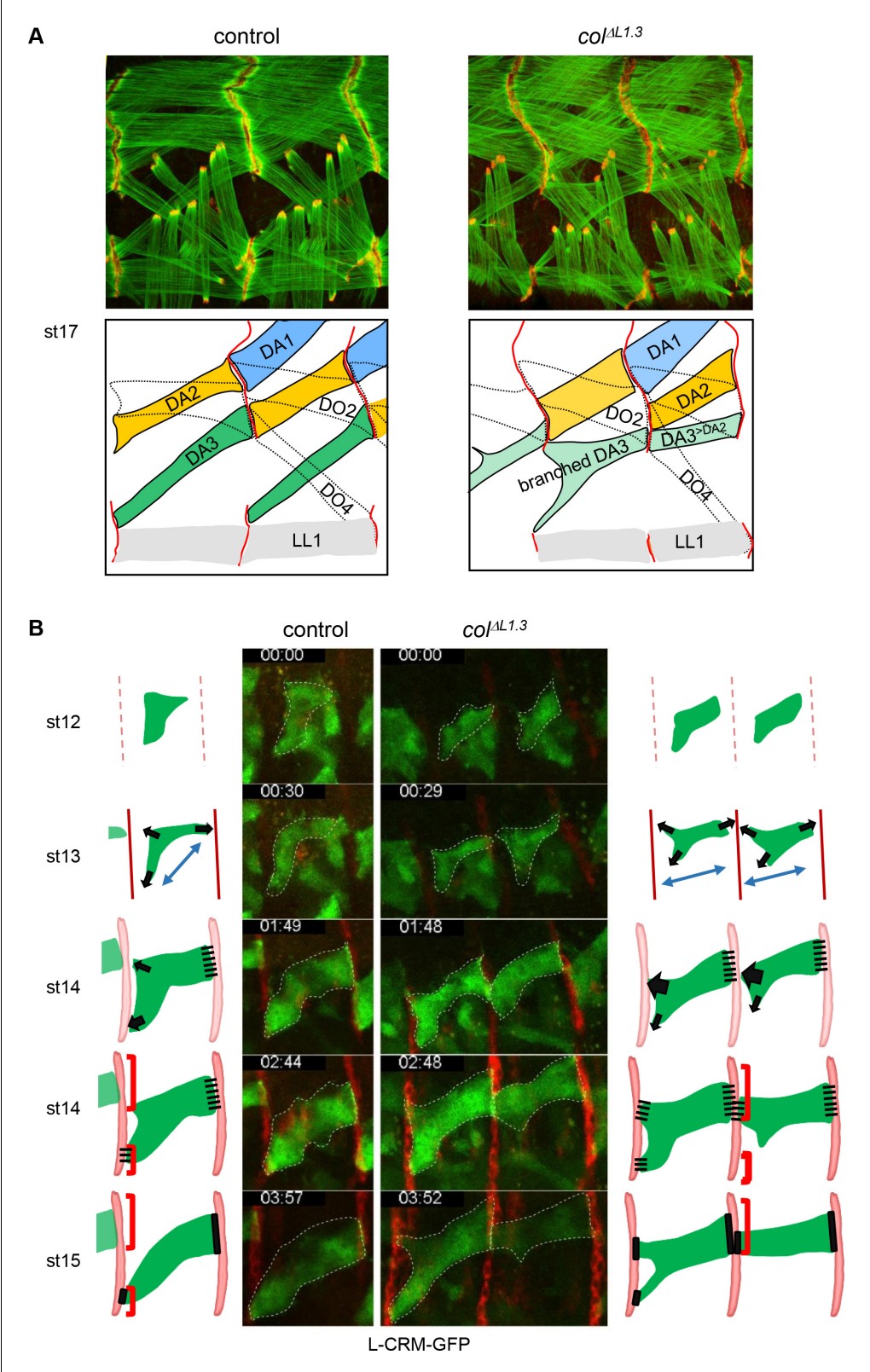

**Figure 4.** Multistep muscle attachment to selected tendon cells. (**A**) F-actin (green) and Ilk-GFP (red) staining of stage 17 control and $col^{\Delta L1.3}$ embryos, showing heterotypic and homotypic muscle attachment sites, respectively. A drawing illustrates muscle matching in control embryos and mismatching in $col^{\Delta L1.3}$ embryos, with DA1, DA2 and DA3 muscles coloured in blue, yellow and green, respectively and attachment sites as red lines. (**B**) Snapshots of live imaging DA3 muscle development, using L-CRM-moeGFP expression (green). Tendon cell precursors express stripe-Gal4; UAS-RFP (red).

*Figure 4 continued on next page*

Figure 4 continued

Embryos were filmed during 4 hr (Sup. *Videos 1* and *2*) and Z sections collected every ~2–2.5 min. The outlines of the developing muscles are schematised for each stage. In both control (left, one segment shown) and *col*<sup>ΔL1.3</sup> embryos (right, two segments shown), the posterior muscle end reaches the intersegmental border first. At stage 14, *wt* DA3 contacts both dorsal and lateral tendon cells at its anterior end. In *col*<sup>ΔL1.3</sup> embryos, the anterior dorsal projections fail to retract, leading to ectopic DA3<sup>>DA2</sup> attachment and branched muscles.

## Muscle attachment: heterotypic adhesion

Further determining how DA3$^{>DA2}$/DA3$^{>DA2}$ homotypic attachment could interfere with the staggered rows pattern of DA muscles, required to visualise at the same time the DA3 and DA2 muscles and tendon cells contours. Vestigial (Vg) is expressed in DA muscles (*Deng et al., 2010*; *Tixier et al., 2010*). We screened the *vg* regulatory landscape and characterised one *vg* CRM active in the DA3 and DA2 (and VL1) muscles, VgM1 (*Figure 5—figure supplement 1*). Expressing together VgM1-mCD8GFP-H2bRFP, L-CRM-moeGFP and stripeGal4:UAS-mCD8RFP, and co-staining of stage 16 control embryos for α-Spectrin, a protein enriched at muscle attachment sites, shows that the posterior edge of DA3 precisely aligns with the anterior edge of DA2 (*Figure 5A*, *Figure 5—figure supplement 1*), suggesting heterotypic attractive cues. This attraction is already observed at stage 14, when the dorsal DA3 and ventral DA2 surfaces contact each other *via* numerous protrusions (*Figure 5A and B*; *Figure 5—figure supplement 2*). Strikingly, at stage 15, the contact zone gets away from intersegmental borders and the DA3/DA3 and DA2/DA2 homotypic connections disappear, to give way to heterotypic DA3/DA2 stable iMAS formation. In *col*$^{ΔL1.3}$ embryos, the initial steps, stages 14 and 15, are similar to *wt*, except for the absence of stable connection of DA3$^{>DA2}$ to lateral tendon cells, as seen by live imaging (*Video 2*). At stage 16, DA3$^{>DA2}$/DA3$^{>DA2}$ and DA3$^{>DA2}$/DA2 iMASs are privileged, while similar to control, no homotypic DA2/DA2 iMAS forms, suggesting homotypic repulsion (*Figure 4*; *Figure 5A*). Together, the data indicate that the precise pattern of iMASs and muscles is contributed by a combination of attractive and, possibly, repulsive cues downstream of muscle iTFs (*Figure 5B*).

## Muscle strength lines in larvae

To investigate the impact of embryonic muscle patterning defects on *Drosophila* larval crawling, we first recorded the fraction of DA3, DA3$^{>DA2}$ and branched muscles in 3$^{rd}$ instar *wt* and *col*$^{ΔLCRM}$ larvae expressing GFP under control of the *Myosin heavy chain* (*Mhc*) promotor region (*Figure 6A*). It conforms to the statistics in late embryos (*Figure 2B*), except for an increased proportion of branched muscles, which could reflect under-evaluation of their number in embryos, due to threshold detection limit of L-CRM-moeGFP expression in thin fibres (*Figure 6B*). We then examined the pattern of MASs, using scanning electron microscopy (SEM) of dissected larval filets (*Figure 6C*; *Figure 6—figure supplement 1*). In *wt* larvae, as seen in stage 17 embryos (*Figure 4A*), the anterior edge of the DA3 muscle is anchored to a nodal lateral attachment site shared with the LL1 muscle, while its posterior edge aligns with the anterior edge of the DA2 muscle in the next posterior segment. Precise heterotypic DA2/DA1 iMASs are also visible over each intersegmental border. The regular alignment of the DA1, DA2 and DA3 muscles both draws strength lines spanning three adjacent segments, and regular tension surfaces between segments (*Figure 6C*; *Figure 6—figure supplement 1*). In *col*$^{ΔL1.3}$ larvae, the anterior DA3 muscle attachment has shifted from lateral to dorsal to form a DA3$^{>DA2}$/DA3$^{>DA2}$ homotypic iMAS (*Figure 6C*). This leads in turn to the formation of narrow, ectopic DA2/DA2 contacts and the DA1-DA2-DA3 strength line is distorted (*Figure 6—figure supplement 1*). In case of branched DA3 muscles, two strength lines co-exist (*Figure 6C* and *Figure 6—figure supplement 1*). In conclusion,

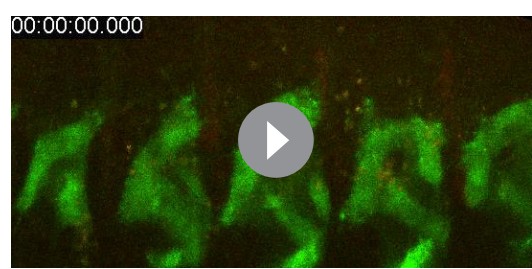

**Video 1.** Live-imaging of *wt* embryos. The DA3 muscle is visualised by L-CRM-moeGFP (green) and tendon cell along the entire intersegmental border by *stripe* Gal4;UASmCD8RFP (red). Embryos were filmed during 4 hr. See also the legend of *Figure 4*.
https://elifesciences.org/articles/57547#video1

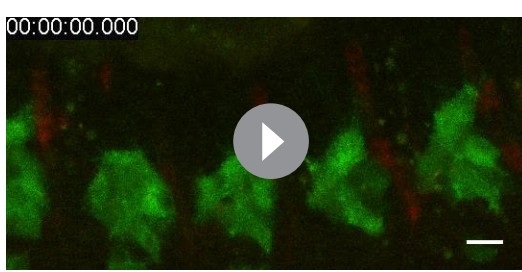

00:00:00.000

**Video 2.** Live-imaging of $col^{\Delta L1.3}$ embryos. The DA3 muscle is visualised by L-CRM-moeGFP (green) and tendon cell along the entire intersegmental border by *stripe* Gal4;UASmCD8RFP (red). Embryos were filmed during 4 hr. See also the legend of **Figure 4**.
https://elifesciences.org/articles/57547#video2

SEM analyses of larval muscles show that loss of DA3 identity leads to ectopic iMASs between several dorsal internal muscles, distortion of the staggered ends architecture of DA muscles and of their alignment between consecutive segments.

## Branched muscles result in subtle locomotion defects

*Drosophila* larval crawling relies upon ordered abdominal body wall muscle contractions (**Heckscher et al., 2012**). The distorted muscle patterns observed in $col^{\Delta L1.3}$ larvae raised the question of whether it impacted on locomotion. To address this question, we compared the locomotion of $+/Df$, $col^{\Delta L1.3}/Df$ and $col^{\Delta L0.5}/Df$ larvae using FIM (FTIR-based Imaging Method) and the FIMTrack software, which allows tracking simultaneously many larvae and quantitatively describing a variety of stereotypic movements (**Risse et al., 2013**; **Risse et al., 2014**; **Risse et al., 2017**). Here, we focused on three parameters: crawling speed, stride length, stride duration. We first recorded the 'walking rate', also called crawling speed, during the first 20 s, after larvae have been dropped on the agarose gel. At that time, larvae engage an 'escape response' corresponding to an active crawling phase (**Figure 7A**). Box-plot graphs (left) show intra-variability for each of the three genotypes. Beyond this variability, we observe, however, that $col^{\Delta L0.5}/Df$ larvae display a significantly reduced crawling speed on average $1.05 \pm 0.034$ mm/sec (n = 108), compared to $1.15 \pm 0.031$ mm/sec for $+/Df$ controls (n = 118), (p=0.03) (**Figure 7A**). To further investigate the origin of this speed reduction, we measured two crawling speed parameters: stride length and stride duration (**Figure 7B–C**). A significantly shorter stride was measured for $col^{\Delta L0.5}/Df$ larvae ($1.08 \pm 0.025$ mm) compared to control $+/Df$ larvae ($1.17 \pm 0.024$ mm), (p=0.008). Furthermore, stride duration was extended, from ($1.09 \pm 0.014$ s) for $+/Df$ larvae to $1.15 \pm 0.017$ s for $col^{\Delta L0.5}/Df$ larvae. For both crawling speed, stride length and stride duration, $col^{\Delta L1.3}/Df$ larvae (n = 112) display intermediate values. Yet, differences with either control or $col^{\Delta L1.3}/Df$ larvae fall below the significance threshold level, suggesting that mis-orientation of the DA3$^{>DA2}$ muscle does not, by itself, significantly impair the efficiency of segment contraction. Since larval crawling integrates information provided by neuronal networks, relayed by synaptic connections between motoneurons (MNs) and muscles, the mobility phenotype of $col^{\Delta L0.5}/Df$ larvae could indicate defects motor innervation of DA3$^{>DA2}$ muscles. The DA3 and DA2 muscles are innervated by the intersegmental nerve (ISN) which fasciculates motor axons reaching dorsal muscles (**Hoang and Chiba, 2001**; **Landgraf and Thor, 2006**). To examine the DA3$^{>DA2}$ and branched DA3 innervation, we used anti-HRP and phalloidin staining to view ISN motoneuron projections and muscles, respectively (**Figure 7—figure supplement 1**). In *wt* embryos, the MN projection which innervates DA3 leaves the ISN ventral to the DA3 position and orients left such that the neuromuscular junction locates to the first anterior third of the muscle (100%; n = 28 segments). In case of a complete DA3$^{>DA2}$ transformation, a MN projection is still observed (100%; n = 15 segments), which leaves the ISN at a more dorsal position than *wt*, reflecting the dorsal shift of the DA3$^{>DA2}$, relative to DA3 muscle. In case of branched muscles, the lower branch is always innervated (100%, n = 30 segments). Only in 20% of the cases, the second, upper branch is also innervated.

These innervation data indicate that fully transformed DA3 muscles are innervated, but only one branch of branched muscles is, in most cases. It remains to be established whether this asymmetric innervation contributes to the reduced crawling speed of $col^{\Delta L0.5}$ larvae.

## Discussion

The stereotyped set of 30 somatic muscles in each abdominal segment which underlies *Drosophila* larval crawling has been thoroughly described many years ago (**Bate, 1990**; **Bate and Rushton, 1993**). One essential aspect laid out at that time was the concept of founder cell (FC), *i.e.*, the

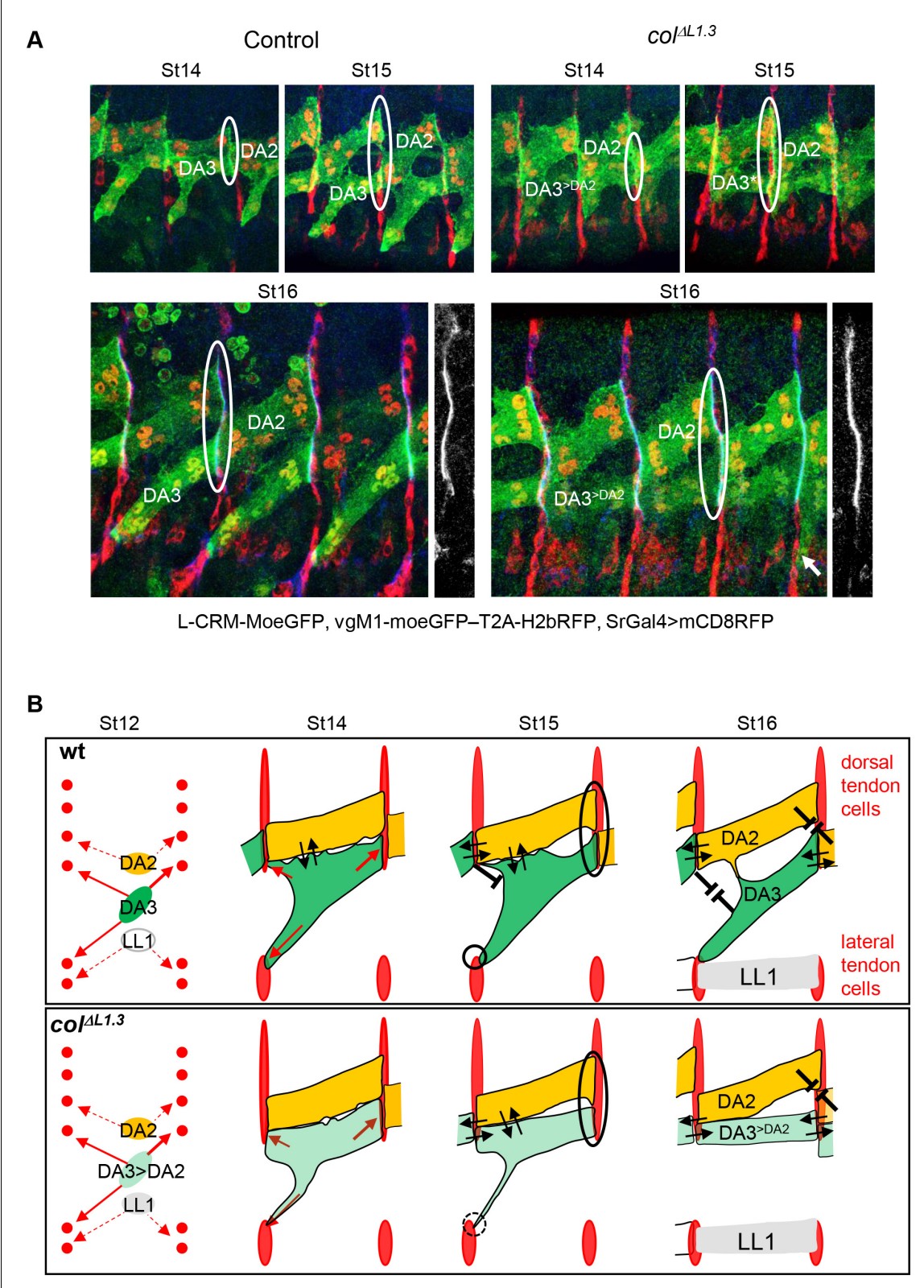

**Figure 5.** DA3 heterotypic and homotypic muscle attachment sites (iMAS) in wt and L-CRM mutants. (**A**) Immunostaining of *wt* and *col*$^{\Delta L1.3}$ VgM1-moeGFP-H2bRFP; srGal4 >mcd8 RFP; L-CRM-moeGFP embryos for GFP (green), RFP (red) and Spectrin (blue). DA3 (DA3$^{>DA2}$) and DA2 muscles are indicated in each panel. At stage 14, the *wt* DA3 and *col*$^{\Delta L1.3}$ DA3$^{>DA2}$ attachment sites partly align over the intersegmental border (circled). Their dorsal surface closely contacts DA2 *via* numerous protrusions; at stage 15, likely because homotypic DA3/DA3 and DA2/DA2 repulsion, these

*Figure 5 continued on next page*

*Figure 5 continued*

protrusions are restricted away from intersegmental borders in *wt* embryos while maintained in *col*$^{\Delta L1.3}$. At stage 16, stable heterotypic DA3/DA2 contacts in *wt*, and both homotypic DA3$^{>DA2}$/DA3$^{>DA2}$ and DA3$^{>DA2}$/DA2 contacts are stabilised. Anti-Spectrin staining is shown in black and white on the right of each panel, to indicate the position of the dorsal iMASs. (B) Schematic drawings of DA muscle development (green) in *wt* and *col*$^{\Delta L1.3}$ embryos; interpreted from data in *Figure 4* and supplementary videos. Arrows indicate attractive cues, broken lines repulsive cues, thin circles attachment initiation, thick circles, attachment stabilisation. DA3 MAS initiation starts earlier at its posterior than anterior end. Homotypic repulsion between muscles of same identity leads to DA3 posterior attachment to lateral tendon cells. Repulsion does not operate upon DA3$^{>DA2}$ transformation, leading to stable homotypic DA3$^{>DA2}$ and DA3$^{>DA2}$/DA2 iMASs.

The online version of this article includes the following figure supplement(s) for figure 5:

**Figure supplement 1.** Identification of a *vg* CRM active in the DA2 and DA3 muscles.
**Figure supplement 2.** Interactions between the developing DA2 and DA3 muscles via dynamic protrusions Immunostaining of a stage 15 VgM1-moeGFP-H2bRFP; srGal4 >mcd8 RFP; L-CRM-moeGFP embryo.

---

assignment to form a distinctive muscle to a single founder myoblast able to recruit other myoblasts by fusion. The morphological identity of each muscle is foreseen by a specific pattern of iTF expression in its FC (*Tixier et al., 2010*; *Frasch, 1999*). Here, we engineered a CRM deletion strategy connecting iTF expression to the larval musculature architecture and locomotion.

## iTF transcription in muscle PCs; redundant CRMs

Some iTFs are transiently transcribed during *Drosophila* muscle development, for example, *Kr* and *nau*, others such as *col*, *ladybird* and *slouch/S59*, at every step of the process (*Dubois et al., 2016*; *Knirr et al., 1999*; *Bataillé et al., 2017*; *Dubois et al., 2007*; *Michelson et al., 1990*; *Jagla et al., 2002*). Reporter analyses indicated that *col* transcription is controlled by two sequentially acting CRMs, of overlapping activity at the PC step, suggesting a handover mechanism between CRMs at this stage (*Enriquez et al., 2012*). However, deletion analyses show that L-CRM activity in PCs is not dependent, but redundant with E-CRM activity, and can be separated from *col* positive autoregulation that is specific to the DA3 lineage. This leads to a new model where iTF code refinement at each step of muscle identity specification, PMC >PC, PC >FC and FC >syncytial nuclei, is driven by a separate CRM. Distribution of the PC-identity information into two CRMs further supports the idea that iTF regulation at the PC stage is nodal to muscle identity specification (*Carmena et al., 1998*; *Dubois et al., 2016*; *Enriquez et al., 2012*; *Jagla et al., 2002*; *Nose et al., 1998*; *Kumar et al., 2015*). Our former analysis started to decrypt the combinatorial control of each dorso-lateral muscle identity, which involves at least eight different iTFs, in addition to Nau/MRF [*Dubois et al., 2007*]. Persistent, low level expression of Col protein in the DT1 and LL1 muscle precursors upon removal of the col autoregulatory module suggests the existence of both positively and negatively acting iTF-responsive elements in this module. Our present analysis concentrated on the DA3 lineage where *col* expression is maintained, whereas *col* is expressed in several PCs. Morphological transformations of other dorso-lateral muscles are observed at variable frequency in *col* protein null mutants (*Enriquez et al., 2010*), suggesting that E-CRM activity provides robustness to the combinatorial control of identity of these muscles and that robustness is likely also contributed by other iTFs (*Dubois et al., 2016*; *Schnorrer and Dickson, 2004*). PC/FC specific CRMs have only been functionally identified for a handful of muscle iTFs (*Rivera et al., 2019*). Computational predictions identified, however, several thousand putative muscle enhancers and uncovered extensive heterogeneity among the combinations of transcription factor binding sites in validated enhancers, beside sites for the core intrinsic muscle regulators Tin, Mef2 and Twi (*Sandmann et al., 2007*; *Gisselbrecht et al., 2013*; *Cusanovich et al., 2018*). Dissecting whether step-specific and redundant/distributed CRM configurations (*Cannavò et al., 2016*; *Hong et al., 2008*; *Frankel et al., 2010*) apply to many muscle iTFs and underlie the progressively refined control of final muscle patterns is a future step.

## Distinctive muscle morphology requires identity reprogramming of fused myoblasts

The process by which iTFs determine the final morphological features of each muscle, is not fully understood. Fusing FCM nuclei generally adopt the iTF protein code of the FC nucleus, while propagation of iTF transcription and activation of realisation genes is muscle lineage-specific (*Boukhatmi et al., 2012*; *Crozatier and Vincent, 1999*; *Knirr et al., 1999*; *Bataillé et al., 2010*;

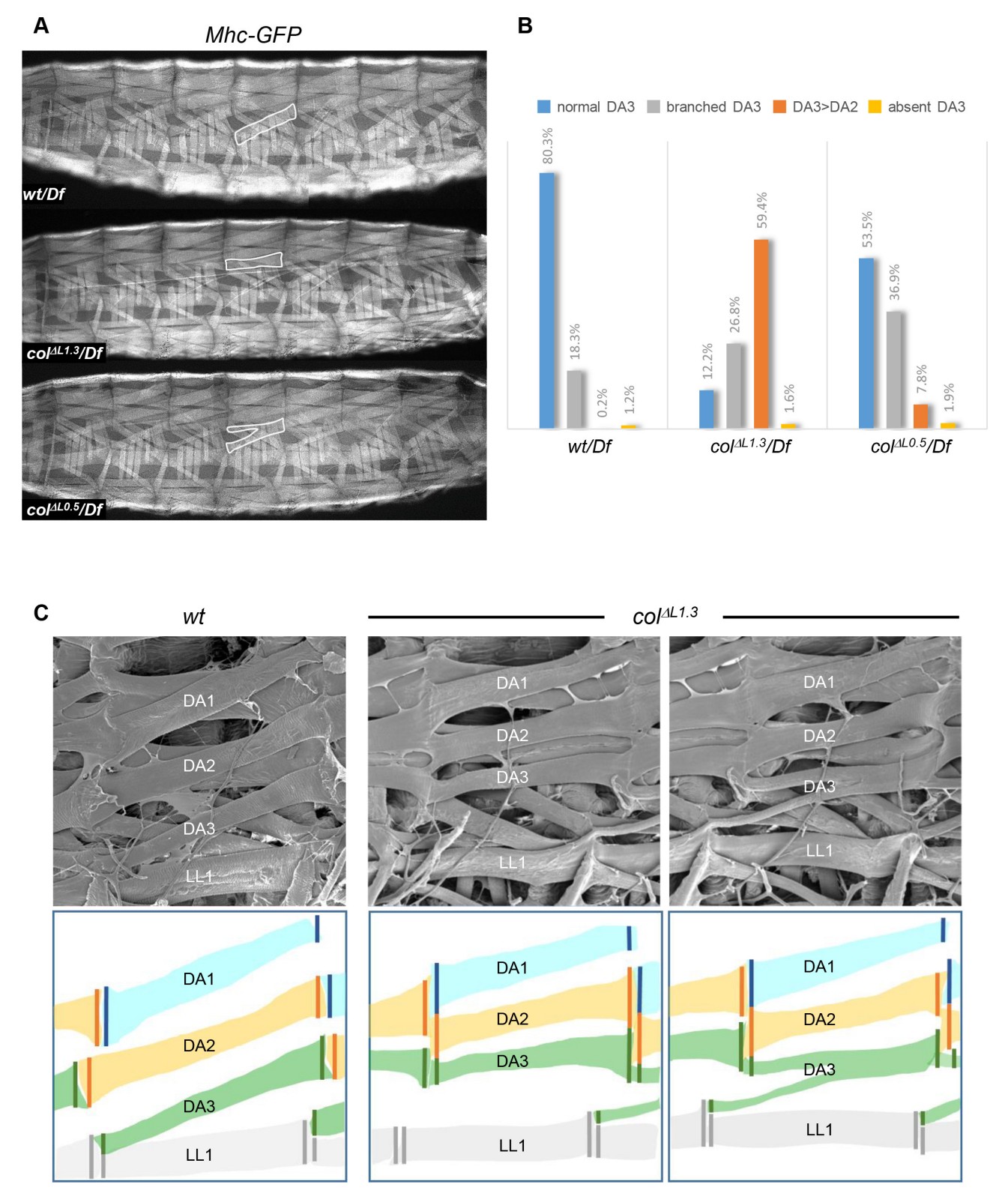

**Figure 6.** Muscle mismatching in L-CRM mutant larvae. (**A**) Mhc-GFP expression in control, *col^ΔL1.3^/Df* and *col^ΔL0.5^/Df* live 3^rd^ instar larvae; lateral views, anterior to the left. Examples of DA3, DA3^>DA2^ and DA3 branched muscles are circled in white. (**B**) Quantification of the relative proportions of DA3, branched DA3, DA3^>DA2^ and absence of DA3 muscles in A1 to A7 segments of *wt/Df*, *col^ΔL1.3^/Df* and *col^ΔL0.5^/Df*; Mhc-GFP larvae; (*wt/Df*, n = 375 segments/27 larvae; *col^ΔL1.3^/Df*, n = 320/23; *col^ΔL0.5^/Df*, n = 264/19). (**C**) Scanning electron microscopy of filleted larvae showing the dorsal and dorso-

*Figure 6 continued on next page*

*Figure 6 continued*

lateral muscles, schematically colour-coded below. In *wt* larvae, the DA muscles are parallel to each other within a segment with precise matching of the DA3/DA2 and DA2/DA1 MASs at each posterior segmental border. In *col*$^{\Delta L1.3}$ larvae, DA3$^{>DA2}$ muscles show homotypic, dorsal MASs and the DA3/LL1 connection is lost. The posterior MASs of branched DA3 is composite.

The online version of this article includes the following figure supplement(s) for figure 6:

**Figure supplement 1.** Scanning electron microscopy of wt and *col*$^{\Delta L1.3}$ internal muscles Larval fillets cut longitudinally along the ventral midline to expose the dorsal and dorso-lateral internal muscles.

*Bataillé et al., 2017*; *Bourgouin et al., 1992*). We have previously shown that Col protein import precedes activation of *col* transcription in fused FCM nuclei, and correlates with the activation of realisation genes, a sequence of events termed syncytial identity reprogramming (*Bataillé et al., 2017*; *Dubois et al., 2007*). Upon loss of *col* transcription in the DA3 FC (*col*$^{\Delta 1.3}$ embryos), there is a complete DA3$^{>DA2}$ transformation. When *col* transcription is maintained (*col*$^{\Delta L0.5}$ embryos), in the DA3 FC but not propagated to other syncytial nuclei, there is an incomplete DA3 transformation into branched muscles. From this, two conclusions can be drawn: 1) Final selection of myotendinous connection sites is intrinsic to FC identity. 2) Identity reprogramming of syncytial nuclei is required for robustness of this selection and precise muscle patterns.

## Identity shifts, a source of branched muscles

Live imaging of lateral-oblique and ventral transverse muscles development distinguished 3 phases of muscle elongation (*Schnorrer and Dickson, 2004*): 1) FC migration, stage 12, ending with the first FC/FCM fusion event and stretching of the muscle precursor along a given axis; 2) bipolar myotube elongation, characterised by the presence of extensive filopodia at both axis ends, in search for attachment sites, stages 13 to 15; 3) maturation of myotendinous attachment, stage 16. The formation of DA3$^{>DA2}$ and branched muscles in CRM mutants recalls a transient exploration by the wt DA3 muscle precursor of both dorsal and lateral tendon cells, a process deviating from the bipolar migration/attachment scheme (*Schweitzer et al., 2010*; *Enriquez et al., 2012*; *Schnorrer and Dickson, 2004*; *Bahri et al., 2009*). Interestingly, ectopic Col expression in the DA2 muscle leads to reciprocal DA2$^{>DA3}$ transformation as well as branched muscles (*Boukhatmi et al., 2012*). This indicates that selection of dorsal *versus* lateral tendon cells is a highly controlled process. A few molecules involved in targeted attachment of subsets of muscles to specific tendon cells have been identified: Kon tiki/Perdido, a single pass transmembrane protein and the PDZ protein DGrip for proper elongation of ventral longitudinal muscles (*Schnorrer et al., 2007*); the ArfGAp protein Git for sensing integrin signaling and halting elongation of Lateral Transverse (LT) muscles once their attachment site has been reached (*Bahri et al., 2009*; *Richier et al., 2018*). Robo/Slit signaling attracts muscles at segmental borders, the Slit ligand being expressed by tendons, and Robo and Robo2 receptors by elongating muscles. slit also acts as a short-range repellent contributing to the collapse of leading-edge filopodia when a muscle reaches the tendon extracellular matrix (*Ordan and Volk, 2015*; *Ordan et al., 2015*). In vivo imaging showed that the DA3 muscle fails to stop at the segment border in slit mutants and sometimes branches (*Ordan et al., 2015*). However, this is not observed in *col*$^{\Delta LCRM}$; the DA3 slit branching pattern is rather similar to that frequently observed in nau mutants, with two posterior attachment sites in place of one (*Dubois et al., 2016*; *Boukhatmi et al., 2012*). Which realisation genes downstream of iTFs are responsible for the precision of DA attachment sites, that is, proper balancing attraction/repulsion cues, should be the focus of future studies.

## Muscle staggered ends; heterotypic *versus* homotypic interactions?

In addition to attaching to tendon cells, it was previously shown that internal muscles attach to each other (*Maartens and Brown, 2015*; *Bate and Rushton, 1993*). Detailed imaging in both embryos and larvae shows that the DA3/DA2 and DA2/DA1 attachment sites precisely match over each segmental border, such that larval DA1, DA2, and DA3 align over three consecutive segments. Recording DA3 and DA3$^{>DA2}$ muscle development by a combination of live imaging and immunostainings

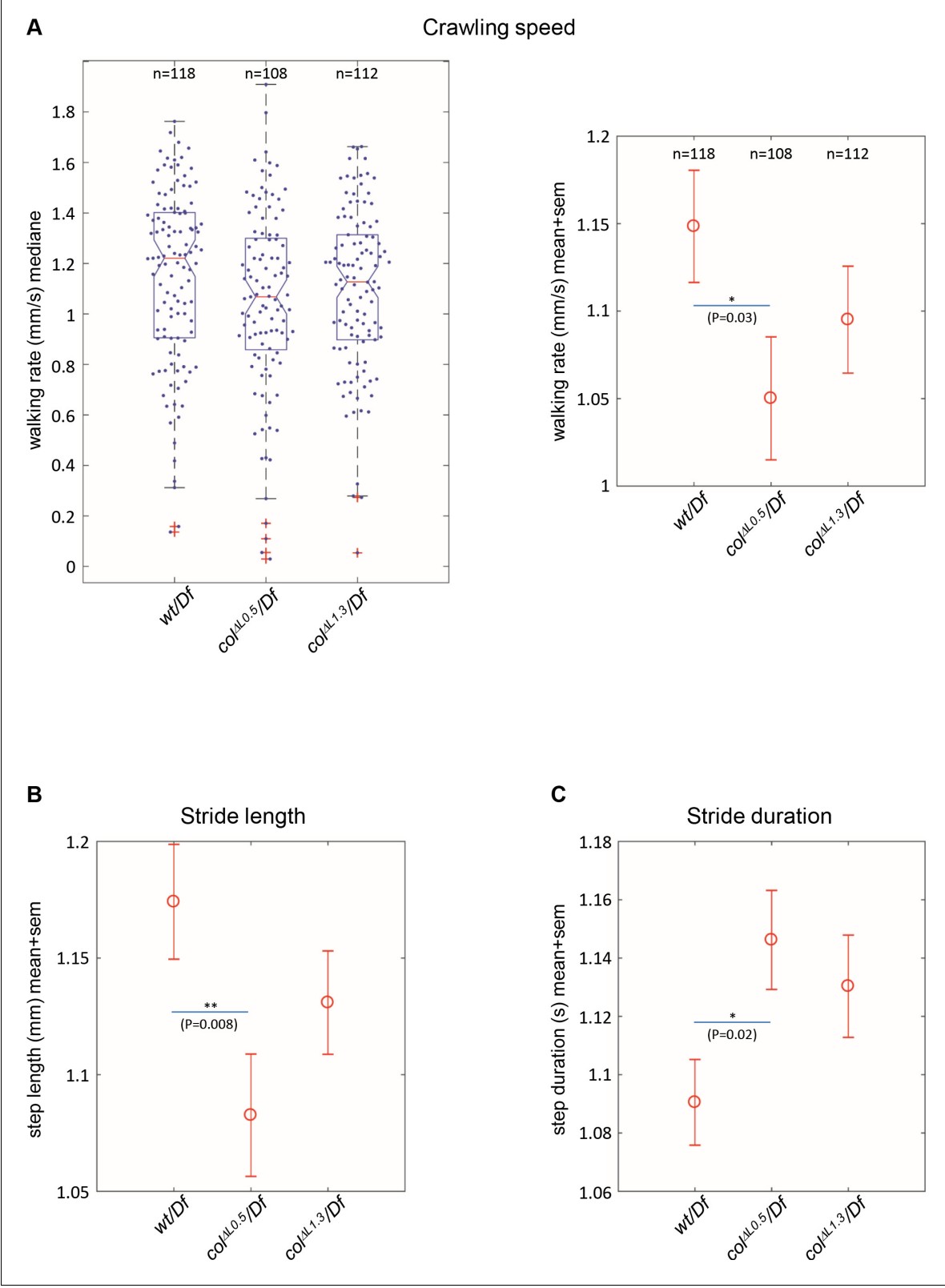

**Figure 7.** Branched muscles result in specific locomotion defects. (**A**) Left, Tukey's diagrams (box-plot graph) showing the walking rate in mm/s (crawling speed) of *wt/Df* (n = 118), *col^{ΔL0.5}/Df* (n = 108) and *col^{ΔL1.3}/Df* (n = 112) larvae. Each point represents the average measurement for one larva, recorded during 20 s. 50% of points are located within the Tukey's diagram. The red line gives the median, the narrowed area, the confidence interval of the median (95%). Right, same data as left, showing the mean speed ± standard error of mean (SEM) for each genotypes. (**B**) Stride length in mm. (**C**)
*Figure 7 continued on next page*

*Figure 7 continued*

Stride duration in s. The number of larvae (n) tested for each genotype is indicated in (**A**). Only significant differences are indicated (*p<0.05 and **p<0.01).

The online version of this article includes the following figure supplement(s) for figure 7:

**Figure supplement 1.** DA muscles innervation in L-CRM mutant larvae.

shows that DA3/DA2 matching in wt embryos results from both heterotypic DA3/DA2 attraction and, possibly, homotypic repulsion. Prior attachment of the posterior DA3 edge to dorsal tendon cells leads to retraction of filopodia issued from the homologous muscle in the next adjacent segment while stabilisation of heterotypic contacts results into DA3/DA2 iMAS formation over the intersegmental border. A similar process results in alignment of DA2 with DA1. Interestingly, DA3$^{>DA2}$ establish both homotypic iMASs, and heterotypic iMASs with DA2, suggesting preserved attraction to tendon cells and partial loss of preferential heterotypic adhesion. Prior posterior attachment was previously observed during development of abdominal adult muscles (*Currie and Bate, 1991*). Whether this temporal sequence is instrumental in the precise matching of muscles over each segmental border and whether competition between muscles for the same tendon cells is also involved remain to be assessed. Cell matching is a widely used process during embryogenesis to construct complex tissue architecture. Selective filopodia adhesion has recently been shown to ensure precise matching between identical cardioblasts and boundaries between different cell identities in the *Drosophila* heart. In this case, homotypic matching is linked to differential expression by each cell type of the adhesion molecules, Fasciclin III and Ten-m (*Zhang et al., 2018*). Transcriptome analyses of specific muscles at different developmental times should allow to identify attractive and, possibly, repulsive molecules, acting in muscle precise matching.

## Branched muscles impact on crawling speed

*Drosophila* larval crawling is a well-suited paradigm to link muscle contraction patterns and locomotor behaviour. Longitudinal, acute and oblique muscles within a larval segment contract together and, as they begin to relax, the contraction is propagated to the next segment, creating a peristaltic wave from tail to head (forward locomotion), or head to tail (backward locomotion) (*Heckscher et al., 2012*). The rhythmic movements of locomotion are part of behavorial routines that facilitate the exploration of an environment. Exploratory routines alternate straight line movement also called 'active crawling phase', with change of direction, the 'reorientation phase' (*Günther et al., 2016*; *Berni et al., 2012*; *Lahiri et al., 2011*). The active larval crawling phase requires an intense, prolonged muscular effort. In this study, we focused on crawling parameters during this phase. The crawling speed during the escape response is not significantly reduced in $col^{\Delta L1.3}/Df$, compared to control +/$Df$ larvae, indicating that muscle contraction is properly controlled and that a mechanical compensation mechanism for the DA3 mis-orientation could occur. However, it is significantly reduced in $col^{\Delta L0.5}/Df$ larvae. This seems paradoxical because the number of DA3$^{>DA2}$ transformed muscles is higher in $col^{\Delta L1.3}/Df$ larvae. $col^{\Delta L0.5}/Df$ larvae present many branched muscles, however. While DA3$^{>DA2}$ are always innervated, only one branch of branched muscles is, most of the time, raising the possibility that branched muscles do not contract properly, or with a gap in time (see *Zarin et al., 2019*). From these different observations, we can conclude: i) single muscle transformations only moderately impact crawling speed, raising the possibility of biomechanical compensation by other muscles; ii) branched muscles could be less efficient than fully transformed muscles. At this point, the reason why, – either mechanic weakness, improper innervation or impaired Ca$^{2+}$ wave propagation, antagonistic force lines upon muscle contraction - may only be object of speculation.

The generation of branched muscles in *Drosophila* identity mutants is one important finding as branched muscle fibres accumulate in humans, following muscle regeneration after damage or in Duchenne muscular dystrophy patients (*Chan and Head, 2011*). It opens the possibility to investigate, in vivo, how physiological properties of branched fibres differ from morphologically normal fibres and associated mechanical instability in an otherwise normal muscle pattern.

# Materials and methods

## Key resources table

| Reagent type (species) or resource | Designation | Source or reference | Identifiers | Additional information |
|---|---|---|---|---|
| Strain, strain background (*Drosophila melanogaster*) | white[1118] | Bloomington *Drosophila* Stock Center | BDSC Cat# 3605, RRID:BDSC_3605 | |
| Strain, strain background (*Drosophila melanogaster*) | Df(2L)BSC429 | Bloomington *Drosophila* Stock Center | BDSC Cat# 24933, RRID:BDSC_24933 | |
| Strain, strain background (*Drosophila melanogaster*) | vasa-cas9VK00027 | Bloomington *Drosophila* Stock Center | BDSC Cat# 51324, RRID:BDSC_51324 | |
| Strain, strain background (*Drosophila melanogaster*) | sr-Gal4 | | | obtained from G. Morata, Madrid, Spain |
| Strain, strain background (*Drosophila melanogaster*) | Col[1] | Our lab | | *Crozatier and Vincent, 1999* |
| Strain, strain background (*Drosophila melanogaster*) | Mi{MIC}knMI15480/SM6a | Bloomington *Drosophila* Stock Center | BDSC Cat# 67516, RRID:BDSC_67516 | MiMic Line |
| Strain, strain background (*Drosophila melanogaster*) | GMR69G03 | Bloomington *Drosophila* Stock Center | BDSC Cat# 39500, RRID:BDSC_39500 | GMR line, GMR located in the Vg gene |
| Strain, strain background (*Drosophila melanogaster*) | GMR69G04 | Bloomington *Drosophila* Stock Center | #46616 | GMR line, GMR in the Vg gene |
| Strain, strain background (*Drosophila melanogaster*) | GMR69G05 | Bloomington *Drosophila* Stock Center | #39501 | GMR line, GMR in the Vg gene |
| Strain, strain background (*Drosophila melanogaster*) | GMR69G06 | Bloomington *Drosophila* Stock Center | #39502 | GMR line, GMR in the Vg gene |
| Strain, strain background (*Drosophila melanogaster*) | GMR69G07 | Bloomington *Drosophila* Stock Center | BDSC Cat# 47956, RRID:BDSC_47956 | GMR line, GMR in the Vg gene |
| Strain, strain background (*Drosophila melanogaster*) | GMR69G08 | Bloomington *Drosophila* Stock Center | BDSC Cat# 46617, RRID:BDSC_46617 | GMR line, GMR in the Vg gene |
| Strain, strain background (*Drosophila melanogaster*) | GMR69G09 | Bloomington *Drosophila* Stock Center | BDSC Cat# 46618 | GMR line, GMR in the Vg gene |
| Strain, strain background (*Drosophila melanogaster*) | GMR69G10 | Bloomington *Drosophila* Stock Center | BDSC Cat# 39503 | GMR line, GMR in the Vg gene |

*Continued on next page*

*Continued*

| Reagent type (species) or resource | Designation | Source or reference | Identifiers | Additional information |
|---|---|---|---|---|
| Strain, strain background (*Drosophila melanogaster*) | GMR69G12 | Bloomington *Drosophila* Stock Center | BDSC Cat# 46619 | GMR line, GMR in the Vg gene |
| Strain, strain background (*Drosophila melanogaster*) | GMR69G03 | Bloomington *Drosophila* Stock Center | BDSC Cat# 46620, RRID:BDSC_46620 | GMR line, GMR in the Vg gene |
| Strain, strain background (*Drosophila melanogaster*) | GMR12A09 | Bloomington *Drosophila* Stock Center | BDSC Cat# 47319 | GMR line, GMR in the Kn gene |
| Strain, strain background (*Drosophila melanogaster*) | GMR12G07 | Bloomington *Drosophila* Stock Center | BDSC Cat# 47854, RRID:BDSC_47854 | GMR line, GMR in the Kn gene |
| Strain, strain background (*Drosophila melanogaster*) | GMR12H01 | Bloomington *Drosophila* Stock Center | BDSC Cat# 48528, RRID:BDSC_48528 | GMR line, GMR in the Kn gene |
| Strain, strain background (*Drosophila melanogaster*) | GMR13A11 | Bloomington *Drosophila* Stock Center | BDSC Cat# 49248 | GMR line, GMR in the Kn gene |
| Strain, strain background (*Drosophila melanogaster*) | GMR13B06 | Bloomington *Drosophila* Stock Center | BDSC Cat# 48544, RRID:BDSC_48544 | GMR line, GMR in the Kn gene |
| Strain, strain background (*Drosophila melanogaster*) | GMR13B08 | Bloomington *Drosophila* Stock Center | BDSC Cat# 48546, RRID:BDSC_48546 | GMR line, GMR in the Kn gene |
| Strain, strain background (*Drosophila melanogaster*) | GMR13C09 | Bloomington *Drosophila* Stock Center | BDSC Cat# 48555, RRID:BDSC_48555 | GMR line, GMR in the Kn gene |
| Strain, strain background (*Drosophila melanogaster*) | GMR13C11 | Bloomington *Drosophila* Stock Center | BDSC Cat# 48556, RRID:BDSC_48556 | GMR line, GMR in the Kn gene |
| Strain, strain background (*Drosophila melanogaster*) | GMR13F08 | Bloomington *Drosophila* Stock Center | BDSC Cat# 48576, RRID:BDSC_48576 | GMR line, GMR in the Kn gene |
| Strain, strain background (*Drosophila melanogaster*) | GMR13F10 | Bloomington *Drosophila* Stock Center | BDSC Cat# 48578, RRID:BDSC_48578 | GMR line, GMR in the Kn gene |
| Strain, strain background (*Drosophila melanogaster*) | GMR47D05 | Bloomington *Drosophila* Stock Center | BDSC Cat# 47605, RRID:BDSC_47605 | GMR line, GMR in the Kn gene |
| Strain, strain background (*Drosophila melanogaster*) | GMR46H09 | Bloomington *Drosophila* Stock Center | BDSC Cat# 54712, RRID:BDSC_54712 | GMR line, GMR in the Kn gene |
| Antibody | anti-col (Mouse monoclonal) | Our lab | | 1:50 *Krzemień et al., 2007* |

*Continued on next page*

Continued

| Reagent type (species) or resource | Designation | Source or reference | Identifiers | Additional information |
|---|---|---|---|---|
| Antibody | anti-LacZ (Mouse monoclonal) | Promega | Promega Cat# Z3781, RRID:AB_430877 | 1:1000 |
| Antibody | anti-spectrin (Mouse monoclonal) | Hybridoma Bank | DSHB Cat# 3A9 (323 or M10-2), RRID:AB_528473 | 1:200 |
| Antibody | anti-GFP (Rabbit polyclonal) | Biolabs | Torrey Pines Biolabs Cat# TP401 071519, RRID:AB_10013661 | 1:1000 |
| Antibody | anti-GFP (Chicken polyclonal) | Abcam | Abcam Cat# ab13970, RRID:AB_300798 | 1:500 |
| Antibody | Phalloidin-texas red | Thermofisher Scientific | Cat# Cat#T7471 | 1:500 |
| Antibody | Alexa fluor antibodies 488, 555 and 647 | Molecular probes | | 1:300 |
| Antibody | Alexa fluor 594 anti HRP | Jackson Immunological research | Jackson ImmunoResearch Labs Cat# 123-585-021, RRID:AB_2338966 | 1:300 |
| Antibody | Alexa fluor phalloidin | Thermofisher Scientific | Thermo Fisher Scientific Cat# A12381, RRID:AB_2315633 | 1:500 |
| Antibody | Biotinylated goat anti-mouse | Vector Laboratories | Vector Laboratories Cat# BA-9200, RRID:AB_2336171 | 1:2000 |
| Software, algorithm | ImageJ | *Schneider et al., 2012* | ImageJ, RRID:SCR_003070 | https://imagej.nih.gov/ij/ |
| Software, algorithm | FIMTrack | *Risse et al., 2014* | | https://www.uni-muenster.de/PRIA/en/FIM/ |
| Software, algorithm | MATLAB | MathWorks | MATLAB, RRID:SCR_001622 | https://www.mathworks.com/products/matlab-online.html |
| Other | FISH probes labelled with Quasar dye 670 (col) | Biosearch Technologies and this study | | Kn first intron |
| Other | FISH probes labelled with Quasar dye 570 (nau) | Biosearch Technologies and this study | | Nau first and third introns |

## Fly strains

All *Drosophila melanogaster* stocks and genetic crosses were grown using standard medium at 25°C. The strains used were *white[1118]*, *col*LCRM 4–0.9 (*Enriquez et al., 2010*), *col[1]* (*Crozatier and Vincent, 1999*), *sr-Gal4* (obtained from G. Morata, Madrid, Spain). The 12 *kn* and 10 *vg* Janelia-Gal4 lines (GMR) (*Pfeiffer et al., 2008*), *UAS-mcd8RFP*, *Mhc-GFP*, *Df(2L)BSC429*, *kn[MI15480] y[1] w[*]*; *Mi{MIC}kn[MI15480]/SM6a* (BDSC_67516) (*Nagarkar-Jaiswal et al., 2015*), *vasa-cas9[VK00027]* (BDSC_51324), *Ilk-GFP* (*w[1118]*; *P{PTT-GB}Ilk[ZCL3111]*) (BDSC_6831), lines were provided by the Bloomington *Drosophila* Stock Center. The *col[1]* and Df(2L)BSC429 strains were balanced using CyO,{*wg[en11]-lacZ*} or CyO,

{dfd-YFP} and homozygous mutant embryos or larvae identified by absence of lacZ or YFP expression, respectively.

## CRM deletions generated by Crispr/Cas9

Genomic col target sites were identified using http://tools.flycrispr.molbio.wisc.edu/targetFinder/ (*Gratz et al., 2014*). Prior to final selection of RNA guides (gRNA) for deletions of col CRMs, genomic PCR and sequencing of DNA from $kn^{MI15480}$ and $vasa\text{-}cas9^{VK00027}$ flies was performed to check for polymorphisms in the targeted regions. Guides targeting E-CRM and L-CRM were inserted in the pCFD4: U6:3-gRNA vector (Addgene n°: 49411) as described (*Port et al., 2014*); (see http://www.crisprflydesign.org/wp-content/uploads/2014/06/Cloning-with-pCFD4.pdf). All guides were verified by sequencing. The sequences of the oligonucleotides used to construct each gRNA expression plasmid are given in *Figure 1—figure supplements 1* and *2*. To delete the core region of L-CRM, vasa-cas9 embryos were microinjected with gRNAs in pCFD4 (200 ng/μl). To delete the E-CRM, $kn^{MI15480}$ embryos were injected with gRNA in pCFD4 (150 ng/μl) and pAct-Cas9U6 (400 ng/μl). Each adult hatched from an injected embryo was crossed to the balancer stock $sna^{Sco}$/CyO, {$wg^{en11}$-LacZ} and 100–200 F1 fly were individually tested for either col CRM deletion by PCR on genomic DNA. A pre-screening for E-CRM deletion was based on the loss of yellow carried by Mimic $kn^{MI15480}$.

## Reporter constructs

The yellow intron (yi), FlyBase ID #FBgn0004034 (position: 356918–359616) was inserted in the lacZ coding region between aa (Tyr 952) and aa (Ser 953) by standard PCR-based cloning position. The resulting fragment was cloned downstream of L-CRM inserted in a pAttB vector, and micro-injected in embryos for chromosomal insertion at position 68A4. VgM1-moeGFP was constructed by PCR amplification of the GMR69G04 and GMR69G05 overlap. The 1.4 kb amplicon (named VgM1) was inserted upstream of moeGFP to generate the pAttB VgM1-moeGFP construct. It was inserted at position 68A4 on the third chromosome.

## Immunohistochemistry

Antibody staining, in situ hybridisation with intronic probes and phalloidin staining were as described previously (*Dubois et al., 2007*). Primary antibodies were: mouse anti-Col (1/50; *Boukhatmi et al., 2012*; *Dubois et al., 2007*), anti-LacZ (1/1000; Promega), mouse anti α-Spectrin (1/200; Hybridoma Bank), rabbit anti-GFP (1/1000; Torrey Pines Biolabs), chicken anti-GFP (1/500; Abcam), Phalloidin-Texas RedX (1/500; Thermofisher Scientific). Secondary antibodies were: Alexa Fluor 488-, 555- and 647- conjugated antibodies (1/300; Molecular Probes) and biotinylated goat anti-mouse (1/2000; Vector Laboratories).

## Motoneurons and muscles visualisation

To both visualise motoneuron axonal pathways and muscles, fillets of control and $col^{\Delta L1.3}$ third instar larvae were incubated overnight with Alexa 594-conjugated anti-HRP (1/300; Jackson Immunological Research) and Alexa Fluor 488 Phalloidin (1/500; Thermofisher Scientific), at 4°C. To prepare fillets, third instar larvae placed in myorelaxant buffer (*Yalgin et al., 2011*), were cut longitudinally on the ventral side to expose the dorsal and dorso-lateral musculature. Fillets were then fixed 1 hr in 4% formaldehyde and washed in PBT.

## In situ hybridisation

In situ hybridisation with Stellaris RNA FISH probes were done as described by the manufacturer for *Drosophila* embryos (https://www.biosearchtech.com). The FISH probe sets for col and nau were designed using the Stellaris probe designer (https://www.biosearchtech.com/stellarisdesigner) and labelled with Quasar 670 Dye (col) and Quasar 570 Dye (nau) (Stellaris Biosearch Technologies). One set of 48 oligonucleotides was designed against the first col intron to detect primary nuclear transcripts. Another set of 48 oligonucleotides was also designed against the first and third nau introns. When antibody staining and FISH were combined, the standard immuno-histochemistry protocol was performed first, with 1 U/μl of RNase inhibitor from Promega included in all solutions, followed by the FISH protocol. Confocal sections were acquired on Leica SP8 or SPE microscopes at 40x or

63x magnification, 1024/1024 pixel resolution. Images were assembled using ImageJ and Photoshop softwares.

## Quantification of *col* transcription

To quantify the level of *col* nuclear transcripts in FCs and PCs, we calculated the ratio between *col* and *nau* hybridisation signals using intronic probes. Before using *nau* as internal reference we verified that *nau* transcription level is not modified in *col* L-CRM mutants. The same laser parameters were set for all intronic probes and at least five different embryos at each stage 11 and 12 were recorded. Optimal Z stacks were acquired at × 40. ImageJ was used to analyse the data. For each stack, a Sum slices projection was generated. Each region of interest (ROI), corresponding to a DA3 nucleus, was manually drawn, based on Mef-2 immunostaining. The same ROI served to determine the intensity of *nau* and *col* signals on the green and red channels, respectively (*Figure 3—figure supplement 2A*). A threshold was applied to each channel to remove background. Data plots and statistical analyses were performed with Prism 5.0 using unpaired *t*-test.

## Phenotype quantification at embryonic and larval stages

To quantify embryonic phenotypes, L-CRM-moeGFP embryos were immunostained with a primary mouse anti-GFP (1/500) (Roche) and secondary biotinylated goat anti-mouse (1/2000) (VECTASTAIN ABC Kit). Stained embryos were imaged using a Nikon eclipse 80i microscope and a Nikon digital camera DXM 1200C. A minimum 100 A1-A7 abdominal segments of stage 15–16 embryos were analysed for each genotype. (+/Df: n = 127 segments - 16 embryos; $col^{\Delta E}$/Df: n = 170 segments - 23 embryos; $col^{\Delta L1.3}$/Df: n = 190 segments - 27 embryos; $col^{\Delta L0.5}$/Df: n = 103 segments - 13 embryos). To quantify larval phenotypes, wandering L3 larva displaying *Mhc-GFP* reporter line were immobilised between slide and coverslip, and left and right larval sides imaged using Nikon AZ100 Macroscope at 5x magnification. Minimum 260 abdominal segments were analysed for each genotype. (+/Df: n = 375 segments - 27 larvae; $col^{\Delta L1.3}$/Df: n = 320 segments - 23 larvae; $col^{\Delta L0.5}$/Df: n = 264 segments - 19 larvae).

## Live imaging embryonic muscle development

Embryos were bleach dechorionated and stage 12 embryos manually picked, laterally orientated and mounted on a coverslip coated with heptane glue to prevent drift during imaging. A drop of water was placed on the embryos to maintain their survival. Images were collected on a Leica TCS-SP8 confocal using a 25X water immersion lens. Sections were recorded every 130 to 150 s for the wt embryos and every 120 to 160 s for the $col^{\Delta L1.3}$ embryos, and z-stacks collected with optical sections at maximum 1 µm interval. Image processing was performed with Fiji (http://fiji.sc/wiki/index.php/Fiji) and custom programming scripts in Fiji. The z-stacks projections were corrected in x and y dimensions by manual registration using a reference point tracking.

## Scanning electron microscopy (SEM)

To prepare fillets, third instar wild type and homozygous $col^{\Delta L1.3}$ larvae raised at 25°C were dissected in myorelaxant buffer, according to *Gratz et al., 2014*. Larvae were cut longitudinally on the ventral side to preserve and expose the dorsal and dorso-lateral musculature. Fillets were then fixed 1 hr in a 4% formaldehyde/2.5% glutaraldehyde mixture in 1X PBS, washed in water and dehydrated gradually in ethanol. Fillets were dried at the critical point (Leica EM CPD 300 critical point apparatus), covered with a platinum layer (Leica EM MED 020 metalliser) and imaged with a Quanta 250 FEG FEI scanning microscope.

## Behavioral analysis

We conducted locomotion assays by tracking the trajectory of larvae using the FIM method (*Risse et al., 2013*). Wandering third instar larvae were gently picked up with a paintbrush and transferred to an agar plate. The larvae were then videotaped using a digital camera (Baumer VCXG53M); lentille (Kowa LM16HC); infrared filter (IF093SH35.5). Each video containing 5 to 10 larvae per run, on a 1% agarose gel, was recorded at five frames/sec for 20 s. Individual larva were tracked using the FIMTrack software (*Risse et al., 2014*), which provided the position across time of five points regularly spaced along the spine of each animal, from head to tail. Analysis was done by using

MATLAB software. Peristalsis cycles were obtained using the derivative of the spine length (i.e., sum of the distance across successive point along the spine) through time, which provide a time series smoothly oscillating around zero. For each peristalsis cycle, we measured Stride length (centroid displacement across each cycle) and Stride duration (cycle duration). Walking rate was obtained by measuring the distance of the centroid (3$^{rd}$ spine point) across successive frame. These values were averaged for each individual across the 20 s of recording. Statistical comparisons between genotypes were computed using a linear model with GenoT as fixed effect, and individual larva as a statistical unit.

## Acknowledgements

We thank the Bloomington Stock Center for *Drosophila* strains, Matthias Landgraf (Cambridge University) for the gift of anti-HRP, Cristian Pasquaretta (CRCA) for his help in statistical analyses of locomotion behaviour, Patrick Arrufat (CRCA), Philippe Firmin and Claude Nexon (CBD) for their assistance in building the locomotion tracking table, Brice Ronsin, Toulouse RIO Imaging platform and Julien Favier, *Drosophila* embryos microinjection platform. We also thank Hadi Boukhatmi, Alice David, Bruno Monier and Serge Plaza for their critical reading of the manuscript. This work was supported by CNRS, Association Française contre les Myopathies (AFM) Research Grant 21887, ANR grant 13-BSVE2-0010-01 and Centre de Biologie Intégrative, AOCBI2018.

## Additional information

### Funding

| Funder | Grant reference number | Author |
| --- | --- | --- |
| Centre National de la Recherche Scientifique | | Alexandre Carayon<br>Laetitia Bataillé<br>Gaëlle Lebreton<br>Laurence Dubois<br>Aurore Pelletier<br>Yannick Carrier<br>Antoine Wystrach<br>Alain Vincent<br>Jean-Louis Frendo |
| Centre de Biologie Integrative de Toulouse | AOCBI2018 | Jean-Louis Frendo |
| AFM-Téléthon | Research grant 21887 | Alain Vincent |
| Agence Nationale de la Recherche | 13-BSVE2-0010-01 | Alain Vincent |

The funders had no role in study design, data collection and interpretation, or the decision to submit the work for publication.

### Author contributions

Alexandre Carayon, Laurence Dubois, Aurore Pelletier, Data curation, Formal analysis, Methodology; Laetitia Bataillé, Gaëlle Lebreton, Resources, Data curation, Formal analysis, Methodology; Yannick Carrier, Resources, Data curation, Methodology; Antoine Wystrach, Resources, Software, Formal analysis; Alain Vincent, Conceptualization, Data curation, Formal analysis, Funding acquisition, Validation, Investigation, Writing - original draft, Writing - review and editing; Jean-Louis Frendo, Conceptualization, Data curation, Formal analysis, Supervision, Funding acquisition, Validation, Investigation, Methodology, Writing - original draft, Project administration, Writing - review and editing

### Author ORCIDs

Alain Vincent http://orcid.org/0000-0002-2769-7501
Jean-Louis Frendo https://orcid.org/0000-0003-0118-5556

Decision letter and Author response
Decision letter https://doi.org/10.7554/eLife.57547.sa1
Author response https://doi.org/10.7554/eLife.57547.sa2

## Additional files

### Supplementary files

• Transparent reporting form

### Data availability

All data generated or analysed during this study are included in the manuscript and supporting files.

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
