## [Decision Letter]

**Acceptance summary:**

The authors study the gene encoding a muscle identity transcription factor, Collier, through examination of its cis-regulatory modules. They demonstrate how selection of attachment sites in muscle and locomotion are related to muscle identity and through transcriptional reprogramming of syncytial nuclei.

**Decision letter after peer review:**

[Editors’ note: the authors submitted for reconsideration following the decision after peer review. What follows is the decision letter after the first round of review.]

Thank you for submitting your work entitled "Intrinsic control of muscle attachment sites matching" for consideration by *eLife*. Your article has been reviewed by a Senior Editor, and three reviewers. The reviewers have opted to remain anonymous.

Our decision has been reached after consultation between the reviewers. Based on these discussions and the individual reviews below, we regret to inform you that your work will not be considered further for publication in *eLife*.

We do appreciate that some of the results are interesting and novel. However, we are not persuaded that they are a significant advance and of sufficient interest for *eLife* readers at this stage. Also, the manuscript should be re-written so that the conclusions are not over-interpreted. In addition, quantification of the FISH analysis must be provided and compared to the protein distribution and intensity. These are amongst the important experiments suggested below that would make the study more widely relevant. We feel that these could take well over two months to do well. We will be pleased to consider a fresh submission, in due course, that addresses all reviewers' concerns, should the authors wish to consider *eLife*.

Reviewer #1:

The manuscript by Carayon et al. addresses an intriguing question of whether the acquisition of muscle identity could impact muscle function and mobility of an organism. To tackle this they generate a series of regulatory region deficiencies that affect transcription of *Drosophila* muscle iTF *col* required for the formation and properties of DA3 muscle. In particular, they generate two viable *col* regulatory regions mutant lines: one in which a complete DA3 to DA2 transformation takes place and another line with deletion of *col* autoregulatory region in which DA3 is not fully transformed to DA2 and has branched morphology.

Interestingly, mutant larvae with branched DA3 muscles display several mobility defects demonstrating that inappropriate acquisition of identity of one muscle could have a deleterious systemic effect. From this perspective, it is a valuable and novel work that merits to be published in *eLife*.

There are a few aspects that require some clarifications:

1) Is the problem of identity propagation observed at stage 14 in *col^ΔL0.5^* context also seen at later stages in DA3 muscle? In other words, is it only a delay in switching on *col* or permanent incapacity to make fused nuclei transcriptionally active for *col*? FISH experiment with intronic *col* probe on stage 16 embryos would help to clarify this point.

2) Some fraction of DA3 muscles in *col^ΔL0.5^* embryos does not display branching phenotype. Could this normal DA3 morphology be associated with a partial or total identity propagation? Could authors document such cases? This would further strengthen the link between identity propagation and branching.

3) Why one nucleus is still able to transcribe *col* and produce Col protein in the context autoregulation is missing. Also, is Col protein expression level in the col-positive nucleus of *col^ΔL0.5^* DA3 similar to that in wt DA3?

4) Authors show that larvae with branched DA3 move slower than larvae with DA3 fully transformed to DA2 indicating that muscle branching has more impact on muscle function than muscle transformation. This could be the case, however other muscle properties and in particular muscle innervation could contribute to mobility phenotype. Some views showing how branched versus fully transformed DA3 muscles are innervated would be of help.

Reviewer #2:

The manuscript of Carayon et al. investigates how the identity of a muscle cell shapes the choice and structure of muscle attachment to its tendon. Previous work from the group has identified specific muscle enhancers in the identity gene collier that controls its expression during the process of founder specification and muscle morphogenesis. This new manuscript adds to this body of work by making specific genomic deletions of these enhancers via Crispr technology. The group subsequently analyzes the phenotypes that result from the selective removal of these enhancers, through confocal imaging of both fixed and live *Drosophila* samples, by SEM and by locomotion assays. These new reagents allow the investigators to map the changes in the patterns of collier expression, assess the contribution of each enhancer to collier expression, and assess one important aspect of muscle identity – the attachment to the tendon cells. Of the key conclusions, they find that (1) as expected from previously published work in the field, FC identity controls selection and development of muscle attachment to the tendon; (2) the robustness of the muscle pattern depends on the reprogramming of myonuclei after fusion (as suggested by earlier published work); (3) the morphogenesis of the tendon-muscle interaction requires several steps in the process, that when altered, leads to the development of branched muscles; and (4) these alterations in muscle attachment (branched muscles) can lead to defects in muscle activity as measured by locomotion of the organism.

Overall, this work is solid. However, the issue for me becomes whether this work provides significant enough advancement for the muscle biology field to merit publication in *eLife*. Unfortunately, I believe that this manuscript would be better suited for a more specialized journal. To merit consideration in *eLife*, one would want to see, for example, how collier regulates the dynamic attachment process that is described.

Reviewer #3:

In this manuscript, Alexander Carayon et al. analyze the consequences of deletion of transcription cis-regulatory modules of a single muscle identity Transcription Factor (iTF), Collier (Col), which in *Drosophila* embryo regulates the identity of muscle DA3. Previously the authors characterized 3 cis-regulatory regions of the Col gene, responsible for early, late, as well as autoregulatory transcriptional control of Col gene in DA3 muscle. They now have deleted each of these sites and ask how it affects Col transcription in the DA3 muscle, and what would be the outcome in terms of muscle morphology, attachment, and larval movement. They show that the CRM for late expression of Col (and to some extent the autoregulatory CRM) are essential for providing the DA3 identity since in its absence, this muscle is either abnormal or partially transformed into DA2 muscle. Although the contribution of Col to the identity of the DA3 had been reported previously, the present study adds important information regarding the functional contribution of the different Col regulatory regions to the muscle identity, attachment sites, and overall larval movement. These results are clean and convincing.

Essential revisions:

1) Overinterpretation of the data: in the abstract, the authors write that " We show that both selection of muscle attachment sites and muscle/muscle matching is intrinsic to muscle identity and requires transcriptional reprogramming of syncytial nuclei". Whereas they indeed show that selection of muscle attachment sites is intrinsic to muscle identity, there is no direct evidence that it requires transcriptional reprogramming of syncytial nuclei.

2) The Col protein, as well as FISH analysis in the *col^ΔL1.3^* or *col^ΔL0.5^* are both intriguing because they indicate that Col transcription is retained in a single nucleus (presumably the founder nucleus) within the entire syncytium of DA3. Do the authors have an explanation as to why the Col protein, being produced in the syncytial cytoplasm, does not translocate to the neighboring nuclei?

3) It will be also informative to demonstrate where this nucleus resides relative to the other nuclei, and whether this impacts on the outcome in terms of the observed attachment sites. In Figure 3, the authors should show the other nuclei of the DA3 muscle in each of the mutants. This can be easily achieved by co-staining the muscles with GFP-NLS driven by col-Gal4 driver.

4) Why the transformation of DA3 to DA2 is relatively rare? Could it be that additional iTFs exist in the DA3 muscle?

5) Do the authors have any idea as to why the *col^ΔECRM^* is homozygous female sterile. Is it due to an off-target effect?

6) Quantification of the FISH: it would be important to add quantification data as to the reduction in the *col* transcripts in the mutants shown in Figure 3.

7) The authors should explain better what they mean by homotypic repulsion.

8) Did the authors check that muscle genes, such as MHC, TpnC, actin etc, are expressed normally and in the mutant DA3 muscle, and that its innervation occurs ordinarily? Can they rule out that the muscle does not function due to physiological reasons, rather than the aberrant attachments?

[Editors’ note: further revisions were suggested prior to acceptance, as described below.]

Thank you for re-submitting your article "Intrinsic control of muscle attachment sites matching" for consideration by *eLife*. Your article has been reviewed K VijayRaghavan as the Senior Editor, and two reviewers. The reviewers have opted to remain anonymous.

The reviewers have discussed the reviews with one another and the Reviewing Editor has drafted this decision to help you prepare a revised submission.

Summary:

Carayon et al. describe an interesting functional analysis of distinct cis-regulatory modules (CRMs) controlling the expression of the collier (col) gene in a single DA3 muscle. Previous analysis from the Vincent lab identified a critical role for *col* in providing the identity of the DA3 muscle. In the present study, the authors analyzed the functional contribution of specific CRMs for the temporal *col* expression and function, in the DA3 muscle, following their specific deletion. The analysis revealed that deleting the Late CRM (L-CRM, *col^ΔL^*), which contains both Mef2 and Twi binding sites, as well as an autoregulatory Col binding sequences, led to the transformation of the DA3 into the DA2 muscle. Deleting a smaller region within this sequence responsible only for the autoregulatory Col activity led to abnormal split muscle DA3. The authors show that whereas in *col^ΔL^* DA3 muscle the entire transcription of *col* is undetectable at stage 14, in fused multinucleated DA3 muscle, the deletion of only the auto-regulatory Col sequences led to partial *col* transcription. Further image analysis of live embryos indicated the sequence of events leading to the split DA3 phenotype. The authors speculate that Col late expression in all the nuclei of DA3 is required for proper detachment of DA3 from the DA2 muscle at its dorsal site and further attraction to the LL1 attachment site. Additional analysis was performed to extract the crawling behavior of the mutant larvae. This analysis implied that the transformation of DA3 to DA2 did not affect significantly the larval step length, however, larvae with split DA3 did show a change in stride length, which was also correlated with aberrant innervation of the split muscle.

Essential revisions:

Overall, the results regarding the functional contribution of the distinct CRMs are convincing and nicely presented. However, the conclusions regarding muscle-muscle repulsion and the model explaining the sequence of events leading to DA3 transformation are less convincing. The link to larvae crawling behaviour is also puzzling due to the stronger phenotype obtained by partial deletion of Col, relative to its complete deletion, and also due to the effect on muscle innervation. These could be discussed in the 'Discussion' section.

1) Figure 2B – the numbers of embryos analyzed is not indicated.

2) Figure 3B – in *col^ΔL0.5^* st 14 there are extra nuclei labeled below the DA3 muscle. These are not stained in the control. What are these?

3) Figure 3C – why does Col protein not label all nuclei in the control DA3? Why is the green label stronger in *col^ΔL0.5^* relative to *col^ΔL1.3^*.

4) Figure 3D – The quantification of transcription at stage 14 is missing. We are puzzled by the question of why late CRM is not active when the autoregulatory is missing. Mef2 should be still active in all nuclei, should it not?

5) Figure 3D – there is no information of how the fluorescence quantification was done. Did the authors perform background subtraction? Did the authors check whether *nau* transcription is affected or not by *col* depletion? They should provide images and quantification of *nau* FISH as well or give clear reasons of why these could be future work if they are unable to do this now.

6) Figure 5 – The authors describe numerous membrane protrusions, which we do not see in the image provided, and no quantification of these is described. No information regarding homotypic DA3/DA3 and DA2/DA2 is provided. Are these real homotypic adhesions? Again, please give clear reasons for why these could be future work if they are unable to do further experiments now.

7) Figure 5 – The images are very nice, however, we do not know about repulsion or attraction. It is speculation. There could be other explanations. Please tone down and discuss various possibilities

The suggested experiments can be done speedily, assuming that the FISH data are already available. In addition, they should tone down the sentences regarding attraction/repulsion because they do not bring strong evidence for that, other than the live imaging, which is nice but descriptive.

---

## [Author Response]

[Editors’ note: the authors resubmitted a revised version of the paper for consideration. What follows is the authors’ response to the first round of review.]

Reviewer #1:[…]There are a few aspects that require some clarifications:1) Is the problem of identity propagation observed at stage 14 in col^ΔL0.5^ context also seen at later stages in DA3 muscle? In other words, is it only a delay in switching on col or permanent incapacity to make fused nuclei transcriptionally active for col? FISH experiment with intronic col probe on stage 16 embryos would help to clarify this point.

The suggested experiment has now been done and the results are presented in a supplementary figure (new Figure 3—figure supplement 1) and recorded in Results section subsection “(Re)programming of syncytial nuclei is required for muscle morphological identity”.

Briefly, in wt embryos, *col* transcription in all nuclei of the DA3 muscle is switched off at stage 16. The same complete switch off is observed in *col^ΔL0.5^* embryos, showing that there is no delay in activation of *col* transcription in low Col level conditions, but rather a low number of fused nuclei which is transcriptionally reprogrammed to DA3 identity.

2) Some fraction of DA3 muscles in col^ΔL0.5^ embryos does not display branching phenotype. Could this normal DA3 morphology be associated with a partial or total identity propagation? Could authors document such cases? This would further strengthen the link between identity propagation and branching.

We agree with the reviewer that normal DA3 morphology observed in *col^ΔL0.5^* is likely associated with at least partial identity propagation. A proxy is L-CRM-moeGFP expression (Figure 2A), which depends upon Col levels through its autoregulation site. To more directly link DA3 nuclei reprogramming to normal versus branched DA3 formation would require visualization of *col* transcription when selected muscle attachment sites have been stabilized (see Figure 4B). As discussed above, *col* transcription has already ceased at that stage, precluding this experiment.

3) Why one nucleus is still able to transcribe col and produce Col protein in the context autoregulation is missing. Also, is Col protein expression level in the col-positive nucleus of col^ΔL0.5^ DA3 similar to that in wt DA3?

The question of the transcriptional outcome of the FC relative to final muscle identity is indeed a key question. To address this question, we have chosen to quantify the *col* transcription rather than Col protein levels, using stellaris single RNA probes, because of higher accuracy. Results, now presented in new Figure 3D and in subsection “(Re)programming of syncytial nuclei is required for muscle morphological identity” show that the level of *col* transcription in the DA3 FC is nodal for transcriptional reprogramming of syncytial nuclei and proper muscle identity. This is an important addition to the previous version; thanks to the reviewer.

4) Authors show that larvae with branched DA3 move slower than larvae with DA3 fully transformed to DA2 indicating that muscle branching has more impact on muscle function than muscle transformation. This could be the case, however other muscle properties and in particular muscle innervation could contribute to mobility phenotype. Some views showing how branched versus fully transformed DA3 muscles are innervated would be of help.

We thank the reviewer again for their suggestion. Labelling motoneuron axons using anti-HRP staining, shows that fully transformed DA3^>DA2^ muscles are innervated. However, in most cases, only one branch of branched muscles is (new Figure 7—figure supplement 1). This innervation defect could, in part, explain the impaired locomotion specifically observed in larvae with numerous branched muscles, as now stated in subsection “Branched muscles result in subtle locomotion defects”.

Reviewer #2:[…]Overall, this work is solid. However, the issue for me becomes whether this work provides significant enough advancement for the muscle biology field to merit publication in eLife. Unfortunately, I believe that this manuscript would be better suited for a more specialized journal. To merit consideration in eLife, one would want to see, for example, how collier regulates the dynamic attachment process that is described.

How Collier regulates DA3 muscle attachment and identifying Col targets in this process is certainly a priority next step. We already tested several candidate genes, either via phenotypic analyses (e.g., slit), or expression analyses (e.g., members of the Dip and Dpr families; Orban et al., 2013; Tan et al., 2015), focusing on Col direct targets (de Taffin et al., 2015) and their proposed partners, but without success. There are hundreds of extracellular protein variants and this is a long-standing investment. The syncytial nature of muscles when the attachment process takes place and dynamic reprogramming of syncytial nuclei has been, so far, an obstacle to transcriptome and ChIP-seq analyses of single muscle, in any organism. New toolsare being developed in our lab to profile in parallel several dorsal muscles in wt and mutant embryos, but we feel that this goal is beyond the scope of our present analysis which connects iTF CRM regulation and locomotion.

Our detailed analysis of DA3 muscle morphological development, both by live-imaging with a dedicated reporter and on fixed embryos and larvae (modified Figure 4, Figure 5 and Figure 6), provides a novel, unprecedented view of the muscle attachment site selection and attachment process (modified scheme Figure 5B). Data in the previous version suggested, for the first time, that homotypic repulsion and heterotypic attraction are involved in precise muscle-muscle attachments leading to the exquisite staggered muscles pattern. In the present manuscript, we visualized both the DA2 and DA3 muscles in wt and mutant embryos using a newly identified driver (new Figure 5A) and the data fully support our previous conclusions.

Reviewer #3:[…]Essential revisions:1) Overinterpretation of the data: in the abstract, the authors write that " We show that both selection of muscle attachment sites and muscle/muscle matching is intrinsic to muscle identity and requires transcriptional reprogramming of syncytial nuclei". Whereas they indeed show that selection of muscle attachment sites is intrinsic to muscle identity, there is no direct evidence that it requires transcriptional reprogramming of syncytial nuclei.

We agree with the reviewer that there is only indirect evidence that muscle identity requires transcriptional reprogramming of syncytial nuclei. We toned down the sentence in the Abstract and modified highlight 2, accordingly.

2) The Col protein, as well as FISH analysis in the col^ΔL1.3^ or col^ΔL0.5^ are both intriguing because they indicate that Col transcription is retained in a single nucleus (presumably the founder nucleus) within the entire syncytium of DA3. Do the authors have an explanation as to why the Col protein, being produced in the syncytial cytoplasm, does not translocate to the neighboring nuclei?

Immunostainings show thepresence of more Col protein in syncytial nuclei in *col^ΔL0.5^* than *col^ΔL1.3^* mutants (Figure 3B), and this correlates with the severity of the phenotype, branched muscle versus DA3^>DA2^. We also found this result puzzling. We have now measured the level of *col* transcription in the DA3 FC in wt and each CRM mutant (new Figure 3D). It shows that this level is significantly lower than wt in both *col^ΔL0.5^* and *col^ΔL1.3^* mutants, at stage 12, when there is a single nucleus. Beyond the lower level in *col^ΔL1.^*^3^ than *col^ΔL0.5^* at that stage, what differs between these two mutant strains is the maintenance of *col* transcription in the syncytial muscle precursor, which is only observed in *col^ΔL0.5^* mutants (Figure 3C). We infer from these different results that the uptake of Col protein by newly fused nuclei reflects the level of newly synthesized Col protein from the FC. One hypothesis is that, in conditions of low transcription/low translation, uptake is mainly limited to the FC nucleus, because of diffusion rate of the protein. In both *col^ΔL0.5^* and *col^ΔL1.^*^3^ mutants, and unlike in wt, there is no other source of Col protein than the FC because of the deletion of the Col autoregulation site. We have now revised the text (subsection “(Re)programming of syncytial nuclei is required for muscle morphological identity”) to make this point clearer.

3) It will be also informative to demonstrate where this nucleus resides relative to the other nuclei, and whether this impacts on the outcome in terms of the observed attachment sites. In Figure 3, the authors should show the other nuclei of the DA3 muscle in each of the mutants. This can be easily achieved by co-staining the muscles with GFP-NLS driven by col-Gal4 driver.

This is an interesting question, which we are presently unable to answer. Previous studies of *col* and other genes transcription in the DA3 muscle have led to the conclusion that the FC nucleus occupies a specific, central position at stages 14 and 15, when the muscle is angled shape (Bataillé et al., 2017). The cease of transcription did, however, not allow determining its position at stage 16. The absence of *col* transcription, after stage 12 makes impossible to determine whether the FC nucleus adopts a different position in *col^ΔL1.3^* mutants and no reporter we have tested so far allows to specifically track the DA3 FC nucleus in the DA3 syncytium.

4) Why the transformation of DA3 to DA2 is relatively rare? Could it be that additional iTFs exist in the DA3 muscle?

We do not feel that the transformation of DA3 in DA2 muscle in *col^ΔL1.3^* mutants is rare (more than 85% in the embryos and 60% + plus 27% branched muscles) in larvae. Previous experiments indicated a similar penetrance for *col* null mutant embryos (Enriquez et al., 2012), suggesting that other iTFs may contribute DA3 identity. Other iTFs are indeed known to be expressed or/and required in the DA3 muscle (Tixier et al., 2010; Dubois et al., 2016) including Vestigial see Figure 5—figure supplement 1). We added a sentence in subsection “iTF transcription in muscle PCs; redundant CRMs”.

5) Do the authors have any idea as to why the col^ΔE^ is homozygous female sterile. Is it due to an off-target effect?

We verified that *col^ΔL1.3^* female sterility (abortive oocyte development, data not shown) is not due to an additional mutation at the *col* locus, suggesting indeed that it is due to an off-target effect.

6) Quantification of the FISH: it would be important to add quantification data as to the reduction in the col transcripts in the mutants shown in Figure 3.

We thank the reviewer for their important remark. We have now quantified the *col* transcription level in nuclei of PC and FC cells. Results are presented in new Figure 3D and described (subsection “(Re)programming of syncytial nuclei is required for muscle morphological identity”), and show that the level of *col* transcription in the DA3 FC is nodal for transcriptional reprogramming of syncytial nuclei and proper muscle identity.

7) The authors should explain better what do they mean by homotypic repulsion.

We name homotypic repulsion the fact that two muscles of the same identity cannot adhere to each other and form indirect musclemuscle attachments. Internal muscle-muscle attachments are only heterotypic, i.e., form between muscle of distinct identities, e.g., DA3/DA2 and DA2/DA1. Repulsion between two DA3 muscles in wt embryos is both supported by movies (Video 1) and stainings of fixed embryos, which show that exploring filopodia issued from one DA3 muscle are repelled by the DA3 muscle of the next anterior segment. Repulsion does not operate upon (partial) loss of DA3 identity (Video 2). We extensively rephrased the text to better explain this point in the revised version (subsection “Muscle attachment: tendon attraction and homotypic repulsion”).

8) Did the authors check that muscle genes, such as MHC, TpnC, actin etc, are expressed normally and in the mutant DA3 muscle, and that its innervation occurs ordinarily? Can they rule out that the muscle does not function due to physiological reasons, rather than the aberrant attachments?

As far as we can tell, the sarcomeric structure of transformed or branched DA3 muscles is normal (see the new supplementary Figure 7—figure supplement 1). We have previously established that activation of “generic” muscle differentiation and attachment genes, such as Mhc and Kon are synchronously activated in all muscle nuclei, making unlikely that is controlled by iTFs (Bataille et al., 2017). Following the suggestion of reviewers 3 and 1, we looked at “DA3” innervation in *col^ΔL0.5^* and *col^ΔL1.3^* mutants. Labelling motoneuron axons using anti-HRP staining shows that fully transformed DA3^>DA2^ muscles are innervated. However, in most cases, only one branch of branched muscles is (new Figure 7—figure supplement 1). This innervation defect could, in part, explain the impaired locomotion specifically observed in larvae with numerous branched muscles, as now stated in subsection “Branched muscles result in subtle locomotion defects”.

[Editors’ note: what follows is the authors’ response to the second round of review.]

Essential revisions:Overall, the results regarding the functional contribution of the distinct CRMs are convincing and nicely presented. However, the conclusions regarding muscle-muscle repulsion and the model explaining the sequence of events leading to DA3 transformation are less convincing. The link to larvae crawling behaviour is also puzzling due to the stronger phenotype obtained by partial deletion of Col, relative to its complete deletion, and also due to the effect on muscle innervation. These could be discussed in the 'Discussion' section.

Proper crawling of the larva requires integration of information from neuronal networks, relayed by stereotypic synaptic connections between motorneurons and muscles (Landgraf and Thor, 2006). One possible interpretation of the *col^ΔL0.5^*mutant locomotion phenotype is that muscle innervation of branched muscles – one connection for two branches- may interfere with its timing of contraction. In case of complete DA3 ^>DA2^ muscle transformation in *col^ΔL1.3^*mutant larvae, the transformed muscle is innervated by the ISN and could contract normally. It will be interesting in the next future to profile the transformed and branched muscle activity in crawling (following the study of Zarin et al., *eLife,* 2019), or other movements such as larval rolling.

Possible interpretations are discussed in subsection “Branched muscles impact on crawling speed”.

1) Figure 2B – the numbers of embryos analyzed is not indicated.

The number of embryos analyzed in Figure 2B was indicated in subsection “Phenotype quantification at embryonic and larval stages”. It is now also given in the legend of the Figure 2.

2) Figure 3B – in col^ΔL0.5^ st 14 there are extra nuclei labeled below the DA3 muscle. These are not stained in the control. What are these?

We thank the reviewer for pointing out this “ectopic” expression. It corresponds to nuclei of other muscles which, in addition to DA3/DO5, originate from progenitors transcribing *col* (DO4/LL1 and DT1/DO3). At stage 14, under normal conditions, Col expression is only maintained in the DA3 muscle and barely visible in these other muscles. In *col^ΔL0.5^*mutant embryos where the *col* binding site is removed, low level of Col expression are still detected in the DT1 and LL1, and possibly D05 nuclei. We previously noticed a similar “ectopic” expression of a 2.6-0.9 late *col* CRM reporter where the Col binding site was mutated (unpublished data; Author response image 1), while *col* expression in the DA3 is strongly reduced. Our interpretation is that either mutation *or removal* of the Col binding site removes a binding site for a repressing factor (perhaps competing with Col for binding) contributing to Col repression in lineages other than DA3. We have previously reported that several TFs contribute to the lineage-specific transcription/repression of *col* transcription (Figure 7 in Dubois et al., 2016), and this putative repressor remains to be identified (this has been added in subsection “iTF transcription in muscle PCs; redundant CRMs”). Of note Col expression in stage 14 nuclei is dependent in all muscles upon the presence of Twi and Mef2 binding sites in the late CRM (compare *col^ΔL1.3^*and *col^ΔL0.5^*). This is mentioned in subsection “(Re)programming of syncytial nuclei is required for muscle morphological identity”.

**Author response image 1. respfig1:** LacZ immunostaining of stage 14 embryos, carrying a late *col* CRM reporter (p2. 6-0.9LacZ). Left: The Col binding site is intact. LacZ expression is predominantly detected in the DA3 muscle. Right: The Col binding site is mutated (nucleotides in red). Low level of LacZ expression is detected in the DA3 as well as DT1, DO5 and LL1 muscles.

3) Figure 3C – why does Col protein not label all nuclei in the control DA3? Why is the green label stronger in col^ΔL0.5^ relative to col^ΔL1.3^.

We agree with the reviewer that the green channel may obscure the blue channel in some panels of Figure 3C, for example wt/df stage 14. Author response image 2 is the original image showing only the blue and red channels, which shows that Col protein (blue) is present in all nuclei transcribing *col* (red channel). We chose not to show the blue channel alone in submitted Figure 3C not to overload it.

**Author response image 2. respfig2:** 

Of note, we have previously shown that Col protein is detected in all DA3 nuclei in stage 14 wt embryos. What we observed, however, was a mixture of ON/OFF nuclei for transcription of identity transcription factors and realization genes, suggesting that *col* auto-regulation and activation of downstream genes is Col threshold-dependent (Dubois et al., 2007; Bataillé et al., 2017).In the Figure 3C, we use L-CRM-moeGFP expression to visualize the Col-expressing progenitors and DA3 muscle contours. L-CRM-moeGFP activity is depending on the Twi and Mef2 binding sites, from the progenitor stage (see Figure 1—figure supplement 3 and Dubois et al., 2007), explaining why more GFP accumulates in *col^ΔL0.5^*than *col^ΔL1.3^* mutants.

4) Figure 3D – The quantification of transcription at stage 14 is missing. We are puzzled by the question of why late CRM is not active when the autoregulatory is missing. Mef2 should be still active in all nuclei, should it not?

As shown in Figure 3C and summary (Figure 3E), *col* transcription is no more detected in *col^ΔL1.3^*mutants at stage 14, preventing comparative measurements.

Previous analysis of *col* mutants has shown that Col is required for maintenance of its own transcription in a few tissues where it is expressed, among which the DA3 muscle lineage beyond the FC stage (Crozatier et al., 1999; Dubois et al., 2007). Mef2 and Twist are also required for L-CRM activity (this manuscript, see reply to point 2). Our working model is that Mef2 and Twist early binding to the L-CRM (Sandmann et al., 2007) primes the late *col* CRM for Col autoregulation (subsection “(Re)programming of syncytial nuclei is required for muscle morphological identity”).

5) Figure 3D – there is no information of how the fluorescence quantification was done. Did the authors perform background subtraction? Did the authors check whether nau transcription is affected or not by col depletion? They should provide images and quantification of nau FISH as well or give clear reasons of why these could be future work if they are unable to do this now.

We have now modified subsection “Quantification of *col* transcription “ and provide a new supplementary figure (Figure 3—figure supplement 2) showing that *nau* transcription is not particularly modified in ΔCRM *col* mutants.

6) Figure 5 – The authors describe numerous membrane protrusions, which we do not see in the image provided, and no quantification of these is described. No information regarding homotypic DA3/DA3 and DA2/DA2 is provided. Are these real homotypic adhesions? Again, please give clear reasons for why these could be future work if they are unable to do further experiments now.7) Figure 5 – The images are very nice, however, we do not know about repulsion or attraction. It is speculation. There could be other explanations. Please tone down and discuss various possibilities

Following the reviewer’s suggestions, we now added a supplementary figure (Figure 5—figure supplement 2), stage 15 embryo, to show the membrane protrusions issued from DA muscles and contacting each other.

We think that the accumulation of MoeGFP at the matching surface between the DA2 and DA3 muscles supports our conclusion of heterotypic adhesion. Triple staining for the tendon cell and muscle contours and α-spectrin supports the model of “indirect” muscle-muscle attachment underneath tendon cells schematized in Maartens and Brown, 2014 (Author response image 3). We postulate that stabilization of the posterior, before anterior attachment, a developmental sequence also observed during development of abdominal adult muscles (Currie and Bate, 1991) may be instrumental in the precise matching of muscle-muscle attachments.

Conversion of the heterotypic DA3/DA2 matching into homotypic DA3^>DA2^/DA3^>DA2^ adhesion in *col* mutants suggests that DA3>DA3 homotypic repulsion is alleviated. The alternative possibility that heterotypic adhesion is not privileged over homotypic adhesion is also plausible. Future work aims at identifying surface molecules which could mediate heterotypic adhesion and/or homotypic repulsion, using DA3, DA2 or DA3^>DA2^-specific RNA-seq experiments. From our previous studies one the dynamics or reprograming of naïve myoblast nuclei during syncytial elongation (Bataille et al., 2017) the time window may be relatively short, making the experiments challenging.

The text has been modified to tone down conclusions on these aspects and discuss various possibilities; subsection “Muscle staggered ends; heterotypic *versus* homotypic interactions?”.

**Author response image 3. respfig3:**